# Continent-wide genomic signatures of adaptation to urbanisation in a songbird across Europe

Pablo Salmón [1,2,8✉], Arne Jacobs [2,8], Dag Ahrén[1], Clotilde Biard [3], Niels J. Dingemanse [4], Davide M. Dominoni [2], Barbara Helm [2,7], Max Lundberg[1], Juan Carlos Senar [5], Philipp Sprau[4], Marcel E. Visser [6] & Caroline Isaksson [1✉]

Urbanisation is increasing worldwide, and there is now ample evidence of phenotypic changes in wild organisms in response to this novel environment. Yet, the genetic changes and genomic architecture underlying these adaptations are poorly understood. Here, we genotype 192 great tits (*Parus major*) from nine European cities, each paired with an adjacent rural site, to address this major knowledge gap in our understanding of wildlife urban adaptation. We find that a combination of polygenic allele frequency shifts and recurrent selective sweeps are associated with the adaptation of great tits to urban environments. While haplotypes under selection are rarely shared across urban populations, selective sweeps occur within the same genes, mostly linked to neural function and development. Collectively, we show that urban adaptation in a widespread songbird occurs through unique and shared selective sweeps in a core-set of behaviour-linked genes.

[1] Department of Biology, Lund University, Lund, Sweden. [2] Institute of Biodiversity, Animal Health and Comparative Medicine, University of Glasgow, Glasgow, UK. [3] Sorbonne Université, UPEC, Paris 7, CNRS, INRA, IRD, Institut d'Écologie et des Sciences de l'Environnement de Paris, iEES Paris, F-75005 Paris, France. [4] Department of Biology, Ludwig Maximilians University Munich, Munich, Germany. [5] Museu de Ciències Naturals de Barcelona, Barcelona, Spain. [6] Department of Animal Ecology, Netherlands Institute of Ecology (NIOO-KNAW), Wageningen, Netherlands. [7] Present address: GELIFES - Groningen Institute for Evolutionary Life Sciences, University of Groningen, Groningen, The Netherlands. [8] These authors contributed equally: Pablo Salmón, Arne Jacobs. ✉email: pablo.salmon.saro@gmail.com; Caroline.Isaksson@biol.lu.se

U rban development is rapidly expanding across the globe, and although urbanisation is regarded a major threat for wildlife[1], its potential role as an evolutionary driver of adaptation has not been explored until recently[2–5]. Many species show phenotypic adaptations to the multiple urban challenges, such as higher levels of noise, artificial light at night, air pollution, altered food sources or habitat fragmentation[6]. These adaptations include changes in behaviour, e.g., refs. [7–9], morphology and locomotion, e.g., refs. [10–13] or toxin tolerance[14]. Indeed, there is now evidence that the phenotypic divergence between urban and rural populations may have a genetic basis in some species[14,15], in line with the finding that micro-evolutionary adaptations in natural populations can occur within short timescales, particularly in response to human activities[16–18]. The study of the genetic signals of adaptation could provide important insights into the magnitude of the evolutionary change induced by urbanisation on wildlife. However, the short evolutionary timescale, the dependence of evolution on local factors, and the polygenic nature of many phenotypic traits, make detecting the genomic signals of adaptation difficult[17,19].

The majority of studies on the genomic basis of urban adaptation have either focused on a limited number of markers and genes[3,20] or on a narrow geographical scale, e.g., refs. [4,15,21,22], thereby limiting the inferences that can be made on the consistency of genomic responses to urbanisation[5]. Conversely, the study of multiple populations on a broad geographical scale provides a powerful framework to identify the evolutionary forces shaping the genomic responses and to test the repeatability of urban adaptation[15,23,24]. In particular, comparing across populations enables the distinction between random demographic non-adaptive processes, such as genetic drift, and signatures of natural selection, as selection might be expected to affect the same genomic regions (nucleotides or genes) and/or functional pathways across cities[23,25]. The current evidence on the genomic basis of urban adaptations ranges from high parallelism, involving a single or few genes[15,23], to polygenic, and to putative genetic redundancy, e.g., ref. [22]. This discrepancy might be due to the variable and sometimes local urban selection pressures together with the polygenic nature of many adaptive phenotypic traits, which could lessen the likelihood for shared genetic changes at the nucleotide level[26].

In this study, we present a multi-location analysis of the evolutionary response to urbanisation, using the great tit (*Parus major*), to identify the genetic basis of urban adaptation. The great tit is a widely distributed songbird and a model species in urban, evolutionary and ecological research, e.g., refs. [27–36], with demonstrated phenotypic changes in response to urban environments in several populations[29,34,35,37]. In addition, genomic resources are well developed for this species[38] and it is known that across its European range, the species presents low genetic differentiation[21,39–41]. In order to examine genomic responses to urbanisation on a broad geographical scale, we analyse pairs of urban and rural great tit populations from nine localities across Europe (Fig. 1 and Supplementary Table 1). We combine several complementary approaches to: (1) detect allele frequency shifts across many loci, which will facilitate the exploration of the polygenic aspect in the adaptation to urbanisation; (2) search for selective sweeps in urban populations, which will provide information on population-specific signatures of selection; and (3) identify enriched functional pathways associated with genes putatively under selection in urban populations, which will help us to infer which particular phenotypes, known and unknown, underlie urban adaptation in great tits. Overall, the present study deepens our understanding of the evolutionary drivers and forces shaping the genomic landscape of urban adaptation on a continental scale.

## Results and discussion

**Genetic diversity and population structure across European urban and rural populations.** A total of 192 great tits from the nine paired urban–rural populations were genotyped at 517,603 filtered SNPs, with 10–16 individuals per sampling site (Supplementary Table 1). We quantified the relative degree of urbanisation for each site (urbanisation score: $PC_{urb}$, from principal component analysis, PCA; see "Methods", Fig. 1b, Supplementary Fig. 1 and Supplementary Table 1) to inform our genetic downstream analyses. Population structuring based on 314,351 LD (linkage-disequilibrium)-pruned SNPs (excluding small linkage groups and the Z-chromosome) was overall low across the 18 studied sites (Supplementary Fig. 2), with each of the first two principal components explaining <3% of the overall variation across populations (Fig. 2a and Supplementary Fig. 3a). Thus, we used a UMAP (Uniform Manifold Approximation and Projection) approach to summarise the genetic variation along the first 20 PC axes. The UMAP analysis revealed the presence of distinct genetic clusters for some of the localities, although there was still a strong clustering of individuals from Gothenburg, Munich, Milan and Paris (Fig. 2a and Supplementary Fig. 3b; results were comparable when including the Z-chromosome, Supplementary Fig. 3c). This analysis also suggested that the levels of divergence differ strongly between each urban and adjacent rural population (Fig. 2a and Supplementary Fig. 3b). Furthermore, the population pairs from Glasgow and Lisbon showed the highest levels of divergence, with Lisbon and Glasgow separating along PC1 and PC2, respectively (Supplementary Fig. 3a), and also separating strongly in the UMAP plot (Fig. 2a). This increased divergence of Glasgow and Lisbon, both located in the range edge of the great tit distribution, could be explained by their slightly reduced heterozygosity, particularly in the urban populations (Supplementary Table 1). Indeed, population tree analyses using TreeMix supported the presence of increased drift in both urban populations (Fig. 2b). However, overall, we did not find lower heterozygosity levels in any of the nine-urban compared to the rural populations, which suggests that urban colonisation was in general not associated with significant bottlenecks (see Supplementary Table 1 for details; Wilcoxon test: $W = 30$, $P = 0.377$).

Interestingly, the population structure analysis indicated that multiple urban populations, namely Glasgow, Lisbon, Madrid, Malmö, Milan, and to weaker extent Barcelona and Gothenburg, formed distinct genetic clusters and independent drift units together, in some cases, with their adjacent rural counterparts (Fig. 2a, b). Nonetheless, the TreeMix analysis also suggests the presence of migration events across some populations, including long distance migration events, e.g., from Lisbon to Glasgow (Supplementary Fig. 4b). Although, there was no clear pattern in relation to predominant gene flow between urban populations (Supplementary Fig. 4). In contrast to the mentioned population clusters, we did not detect any particular signs of genetic divergence between populations from Munich and Paris (neither PC or UMAP axis see, e.g., Fig. 2a and Supplementary Fig. 3). Likewise, we detected significant genetic differentiation between all pairwise comparisons ($P < 0.05$; $F_{ST}$ permutation test; see "Methods") except for the urban populations in Munich and Paris (Fig. 2e and Supplementary Table 2). Moreover, the genetic population differentiation ($F_{ST}$) within population pairs (urban vs adjacent rural population), as well as between urban populations (urban vs urban), was on average higher than the differentiation between rural populations (rural vs rural) across Europe (Fig. 2c). These results support the idea that gene flow is generally stronger between rural compared to urban habitats in the nine population pairs, and indicates reduced gene flow between urban and adjacent rural populations. This is further supported by the observed isolation-by-distance patterns, which showed slightly

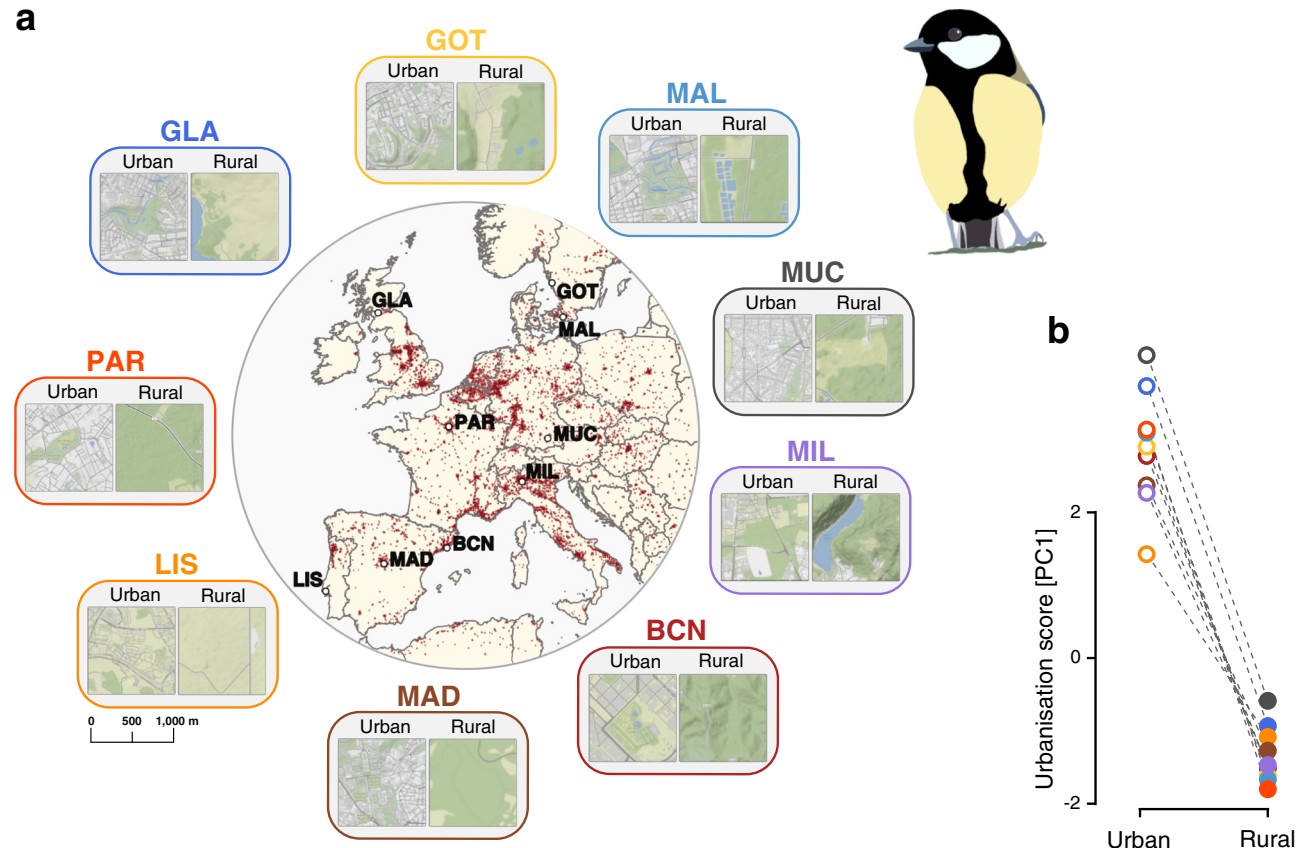

**Fig. 1 Study sampling locations and urbanisation intensity. a** Centre: map of Europe, showing the targeted cities where the sampling of great tits (*Parus major*) was carried out. Red areas indicate the main dense urban areas. The nine inset zooms flanking the central map exemplify the landscape and degree of urbanisation for each of the urban–rural sampling locations. Green shading represents canopy or green areas (i.e., vegetation cover) and roads and buildings are represented in grey. See "Methods", Supplementary Fig. 1 and Supplementary Table 1 for more details. Map data sources: Stamen Design under CC BY 3.0 (https://creativecommons.org/licenses/by/3.0/); data licensed by OpenStreetMap Foundation (OSMF) under ODbL (https://opendatacommons.org/licenses/odbl/). Shapefiles: Natural Earth. **b** Urbanisation scores (principal component, $PC_{urb}$) for all nine urban–rural pairs. In both figures, the coloured circles represent each city (in **b**, urban: open circles; rural: closed circles). BCN Barcelona, GLA Glasgow, GOT Gothenburg, LIS Lisbon, MAD Madrid, MAL Malmö, MIL Milan, MUC Munich, PAR Paris. Great tit illustration made by Pablo Salmón. Source data for **b** is provided as a Source data file.

stronger IBD for urban (Mantel's $r = 0.46$, $P = 0.008$) compared to rural populations ($r = 0.43$, $P = 0.007$; Fig. 2d). Lastly, analysis of the effective migration surfaces confirmed reductions in gene flow in many parts of Europe, compared to neutral expectations, including those in close proximity, e.g., Gothenburg and Malmö, and simultaneously highlighted strong gene flow in Central/Western Europe, i.e., Munich and Paris (Fig. 2e).

Overall, our results reveal weak population structuring across Europe, with slightly increased genetic differentiation between urban populations compared to the more admixed rural populations. Previous results in another European songbird, the blackbird (*Turdus merula*) showed that urban birds likely independently colonised multiple times European cities from forest sources[42]. In our study, the finding of independent clusters for some urban populations also point towards a similar scenario, i.e., an independent and repeated colonisation of urban habitats from largely admixed rural populations. However, the overall weak genetic divergence and the detection of significant migration events across distant populations suggests a possible role of gene flow in the facilitation of urban adaptation, via the spread of adaptive alleles. Still, in our subsequent analyses we treated all our urban–rural population pairs as independent, including Munich and Paris, as selection pressures might differ across cities and result in idiosyncratic genomic responses to selection despite any ongoing gene flow.

**Identification of urbanisation-associated allele frequency shifts**. To identify genomic regions with consistent allele frequency shifts associated with urbanisation, we used two complementary genotype–environment association (GEA) approaches, LFMM (latent-factor mixed models) and an additional Bayesian approach (using BayPass). Testing for GEAs with LFMM (using $PC_{urb}$ as a continuous habitat descriptor, Fig. 1b) revealed 2758 SNPs associated with urbanisation (0.52% of the full SNP dataset, false-discovery rate (FDR) < 1%; Fig. 3a). Urbanisation-associated SNPs were widely distributed across the genome and did not cluster in specific regions. Larger chromosomes ($R^2 = 0.97$) and those with more genes ($R^2 = 0.90$; Fig. 3b) contained more urbanisation-associated SNPs, highlighting the polygenic nature of urban adaptation. A PCA ($PC_{GEA}$), based on all urban-associated SNPs detected by LFMM, clearly separated urban and rural populations along $PC1_{GEA}$ (proportion of variance explained—PVE—by PC1 = 1.98%; Supplementary Fig. 5c), suggesting consistent allele frequencies in those loci across European cities.

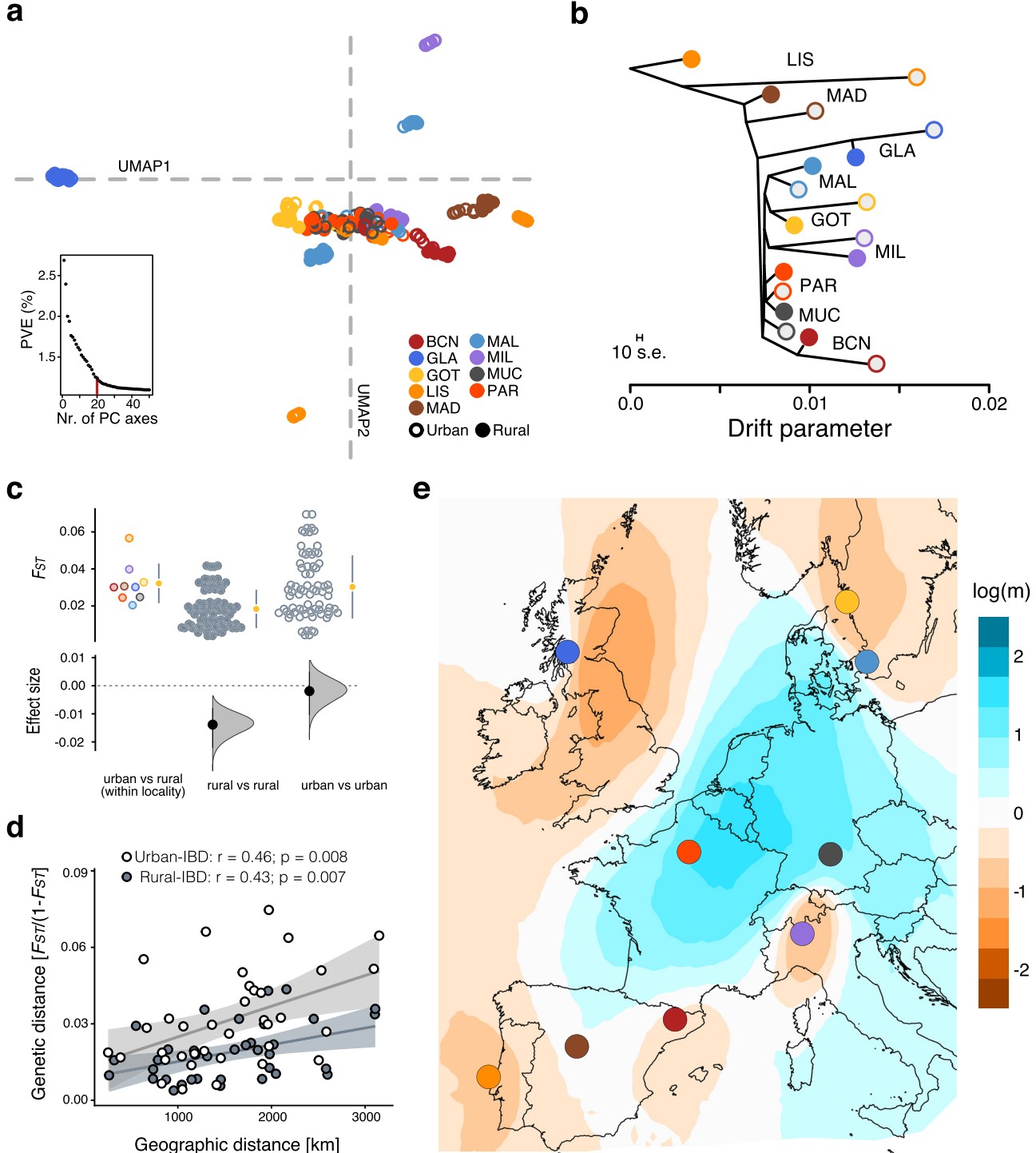

**Fig. 2 Great tit population structure. a** UMAP reduction of the first 20 principal component axes for an LD-pruned SNP dataset, excluding the Z-chromosome and small linkage groups. The insert shows the percent of variance explained (PVE) by each of the first 50 PC axes. The corresponding principal component plots can be found in Supplementary Fig. 2. **b** Population tree generated with TreeMix showing the relationship of urban (open circles) and rural (closed circles) populations from all sampling locations. **c** Comparison of $F_{ST}$ value distributions (mean ± SD) across population and habitats. The effect size plots (mean ± 95% bootstrap CI) show that overall rural populations (rural vs rural) display lower differentiation than the studied population pairs (urban vs rural), but not lower than urban populations (urban vs urban). **d** Isolation-by-distance analyses (Mantel's test) for urban (open circles, light shaded, $n = 72$) and rural (closed circles, dark shaded, $n = 72$) populations separately. Shaded area denotes the 95% CI. **e** Effective migration surfaces for great tits across Europe. Negative log migration rates [log($m$)] depict areas with less gene flow than expected under an isolation-by-distance model, whereas positive migration rates indicate stronger gene flow. Note that migration rates outside the central part of the sampling distribution are generally low, also between closely related cities. BCN Barcelona, GLA Glasgow, GOT Gothenburg, LIS Lisbon, MAD Madrid, MAL Malmö, MIL Milan, MUC Munich, PAR Paris. Colour code is given in panel **a**. Source data are provided as a Source data file.

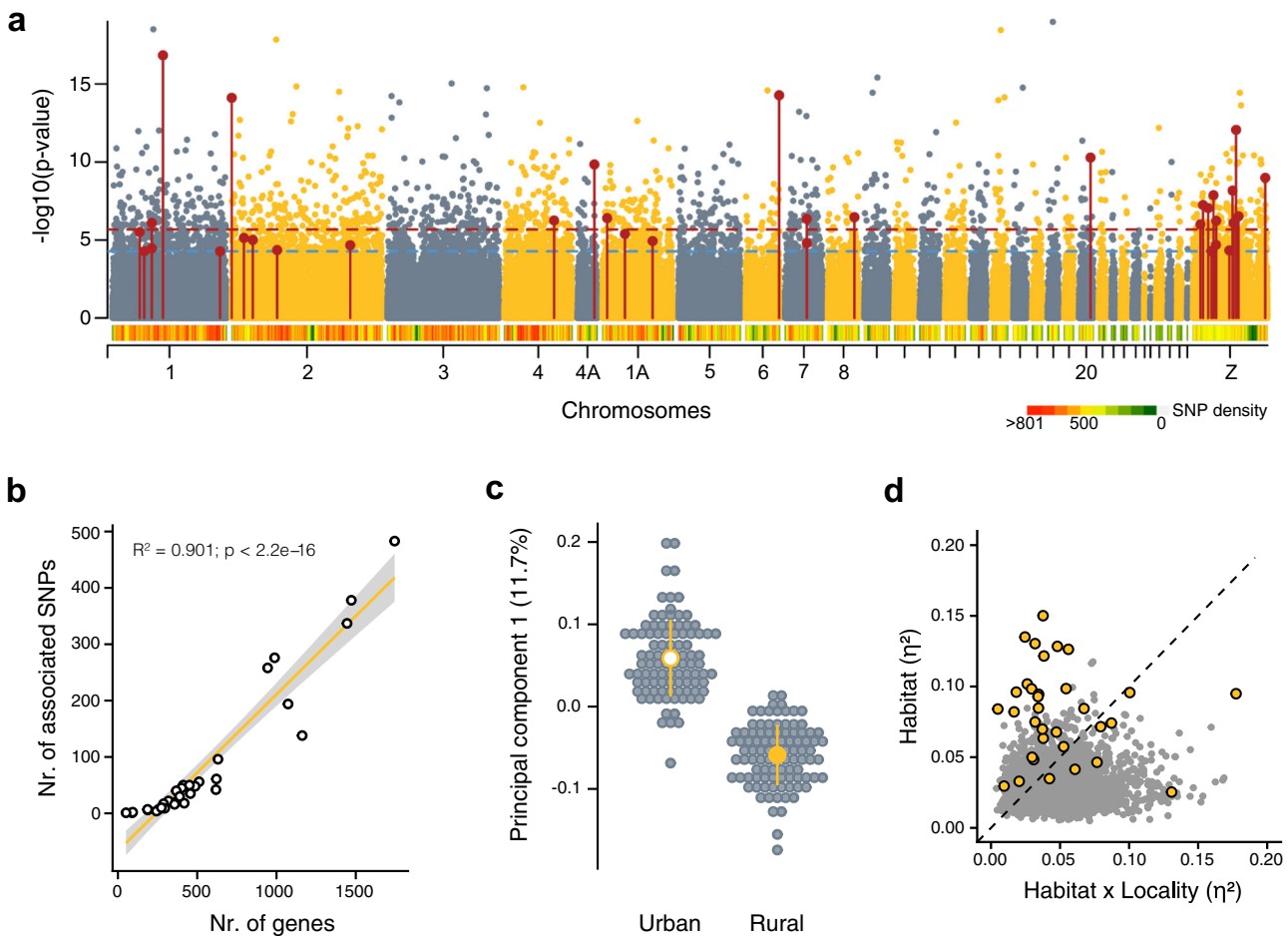

**Fig. 3 Genome-wide association with urbanisation. a** Manhattan plot showing the genotype–urbanisation (PC$_{urb}$) association estimated with LFMM across the genome. The red and blue dotted lines show the 0.1 and 1% FDR significance thresholds, respectively. Red dots highlight "core urbanisation SNPs" that were also identified as urbanisation associated using BayPass (Supplementary Fig. 6). The heatmap below the Manhattan plot highlights the SNP density across the genome. **b** Correlation between the number of urbanisation-associated SNPs (from LFMM) and the number of genes per chromosome (linear model: $F_{1,30} = 284.60$, $P < 2.2 \times 10^{-16}$). Shaded area denotes the 95% CI. **c** Main axis of variation in a PCA$_{GEA}$ based on "core urbanisation SNPs". Grey circles show individuals, and yellow dots and lines show the mean ± SD for urban and rural populations, $n_{urban} = 96$ and $n_{rural} = 96$ independent birds. **d** Effect sizes (partial $\eta^2$) for the effects of "habitat" (urban vs rural) and "habitat × locality" interaction on urban–rural allele frequency, using linear model are shown for all significant LFMM SNPs on the x-axis and y-axis, respectively. "Core urbanisation SNPs" are highlighted in yellow circles. SNPs that lie above the dashed line show strong consistent shifts in allele frequencies across localities, whereas SNPs below the line show more variable allele frequency shifts across localities. Source data are provided as a Source data file.

In contrast, the results obtained by BayPass only identified 70 SNPs strongly associated with urbanisation (Bayes factor ≥ 20; Supplementary Fig. 6a). The lower number of identified SNPs might respond to the stronger population structure correction applied by BayPass, which correlates to some extent with the direction of local adaptation[43]. Of these significant SNPs, 34 were shared with the LFMM analysis (more than expected by chance: $\chi^2 = 31.20$, $P = 2.35 \times 10^{-08}$; Fig. 3a and Supplementary Fig. 6b). These shared SNPs, which we term "core urbanisation SNPs" (Supplementary Table 3), are likely involved in the local adaptation of great tits to urban habitats, and indeed, they more strongly discriminated urban and rural individuals across Europe (PVE by PC1$_{GEA\text{-}shared}$ = 11.7%; Fig. 3c and Supplementary Fig. 7a).

In order to gain a better understanding of the importance of habitat (i.e., urban vs rural) on allele frequency shifts in the detected SNPs, we assessed using univariate linear models, the explanatory power of habitat, locality (city), and their interaction on the direction and strength of allele frequency shifts per-SNP across all paired populations. A significant "habitat" term

indicates consistent allele frequency shifts across cities, particularly when the effect size (partial $\eta^2$) is higher compared to the effect size of a significant "habitat × locality" interaction term, which describes differences in the direction and magnitude of the allele frequency change between cities[44]. Applying these models to each urbanisation-associated SNP from the LFMM model (2758 SNPs), we detected large variation in allele frequency shifts across localities (Fig. 3d and Supplementary Fig. 7b), with a slightly larger proportion of SNPs showing differences in allele frequency shifts across localities ("habitat × locality" $\eta^2$ > "habitat" $\eta^2$). In contrast to this, most of the "core urbanisation SNPs" showed similar allele frequency differences across localities, with 76% of them showing a main effect of the urban habitat (Fig. 3d), and with the same allele increased in frequency in seven or more urban populations (Supplementary Fig. 7c; see Supplementary Fig. 8 for the minor allele frequency trajectory between habitats). We refined this analysis by accounting for the effect of allele frequency correlations between SNPs through the use of the principal component axis from all the LFMM urbanisation-associated SNPs (PC1$_{GEA}$) and the "core urbanisation SNPs"

(PC1$_{GEA\text{-}shared}$, Supplementary Fig. 7d). In both cases, the "habitat" term explained the majority of the total variation in allele frequency divergence ($\eta^2_{PC1\ GEA,\ habitat} = 0.87$, $P < 0.001$; $\eta^2_{PC1\ GEA\text{-}shared,\ habitat} = 0.73$, $P < 0.001$). In comparison, both the effect of "locality", which corresponds to the distinct evolutionary history of the local populations ($\eta^2_{PC1\ GEA,\ locality} = 0.59$, $P < 0.001$; $\eta^2_{PC1\ GEA\text{-}shared,\ locality} = 0.20$, $P < 0.001$), and the interaction of "habitat × locality" ($\eta^2_{PC1\ GEA,\ habitat \times locality} = 0.13$, $P = 0.002$; $\eta^2_{PC1\ GEA\text{-}shared,\ habitat \times locality} = 0.13$, $P = 0.001$), explained much smaller proportions of the total variation (smaller effect size) than the habitat term by itself. In contrast, genetic variation along PC2 and PC3 was mostly inconsistent across localities and likely specific for each city (Supplementary Fig. 7d). Overall, the PCA-based results are in line with those obtained by the per-SNP results, reinforcing the idea of consistent continent-wide responses to urbanisation, in particular regarding the "core urbanisation SNPs".

Together, we show that there is local adaptation to urban habitats in great tits, and that this occurred across Europe through shifts in allele frequency of the same loci. This might have occurred via the standing genetic variation observed for the species, as putatively adaptive alleles were shared across large parts of the species' European distribution[41], or the exchange of adaptive alleles through gene flow across urban populations. Nonetheless, in line with a polygenic basis of adaptation via subtle allele frequency shifts in several loci[26], the detected urban–rural differences were generally small (Supplementary Fig. 9) with only a few SNPs showing relatively strong allele frequency shifts with |ΔAF| (allele frequencies difference between urban and rural) >0.5.

**Signatures of selection in urban populations**. While GEA approaches are powerful tools for identifying even subtle allele frequency shifts associated with local adaptation[43], they have difficulty detecting selective sweeps unique to one or few populations or sweeps of different haplotypes associated with the same gene. Selective sweeps have been associated with urban adaptation in other species, for example, in the case of New York City brown rats (*Rattus norvegicus*) and white-footed mice (*Peromyscus leucopus*)[4,45]. Therefore, we further performed genome-wide scans of differentiation and selective sweep analyses in each of our studied populations to test if these contributed to the urban adaptation in great tits and, importantly, to explore if selective sweeps were population specific or widely shared across Europe.

Since the exchange of genetic material between urban and rural populations, and the selection pressures are likely recent and ongoing in great tits, we opted to use a cross-population statistic that can identify ongoing and recently completed hard and soft selective sweeps (*XP-nSL*)[46]. Using this approach, we found that between 436 and 700 genomic windows (200 kb sliding windows with 50 kb steps, see "Methods"), which clustered into 127–173 wider genomic regions, showed signatures of selection in urban populations (Fig. 4a). Genomic outlier regions had an average size of 355.8 kb ± 251.6 SD, with the largest region (3.05 Mb) detected on chromosome 1 in Glasgow and were widely distributed across the genome with a few distinct population-specific peaks (Fig. 4a). The large number of genomic outlier regions are in line with the high number of significant GEAs across the genome and confirms the inferred polygenic nature of urban adaptation in great tits (Fig. 3a). However, in contrast to the highly concordant allele frequency shifts of GEA loci, *XP-nSL* values were in general weakly correlated across populations (Fig. 4b), and outlier windows were shared across a maximum of five populations (Supplementary Fig. 10). The top three shared windows clustered in a 300 kb region on chromosome 11 (16.05–16.35 Mb) and were detected in Gothenburg, Munich and the Iberian Peninsula

(Barcelona, Madrid and Lisbon), with one partially overlapping outlier window detected in Glasgow. In general, the most pronounced sharing of outlier windows did not seem to occur between the geographically closest populations, as expected for shared genetic variation or adaptive introgression (see Supplementary Fig. 10). For example, Gothenburg and Paris, and Lisbon and Glasgow, showed the highest number of shared outlier windows (53 and 47, respectively). Indeed, the number of shared outlier windows between urban populations was not correlated with their genetic (Mantel's $r = 0.28$, $P = 0.150$) or geographic distance (Mantel's $r = 0.05$, $P = 0.400$), suggesting that shared adaptive genetic variation through introgression and gene flow between neighbouring urban populations likely did not facilitate urban adaptation. However, we cannot fully exclude the role of gene flow as migration was generally strong, even across large distances (Supplementary Fig. 4).

The observed landscape of variation in *XP-nSL* is in agreement with the genomic landscape of genetic differentiation (standardised Z-transformed $F_{ST}$ [$ZF_{ST}$]) between each adjacent urban and rural pair across the same windows (Fig. 4b, Supplementary Figs. 11 and 12). $ZF_{ST}$ outlier windows ($ZF_{ST} > 4$, equivalent to four SD) were largely population specific (Supplementary Fig. 12), suggesting that population-specific selective and demographic processes have partly shaped the genomic landscape[47]. To strengthen our inference of the selective sweeps landscape associated with urban adaptation in our species, we additionally searched for older and completed selective sweeps using the haplotype-based *Rsb* statistic[48]. Similarly to the results obtained using the *XP-nSL* statistic and $ZF_{ST}$, window-based *Rsb* scores were weakly and inconsistently correlated across populations (Fig. 4b and Supplementary Fig. 13). We detected fewer *Rsb* outlier windows (209–538) and clustered genomic outlier regions (39–98; average size: 396.9 kb ± 329.2 SD; max = 3.45 Mb on chromosome 3 in Gothenburg) compared to *XP-nSL*. This result suggests that ongoing/recent sweeps outnumber older and completed selective sweeps in urban great tits. Similar to *XP-nSL*, the *Rsb* outlier windows were only shared across a maximum of five populations and in general widely distributed across the genome (Supplementary Fig. 14).

Overall, the selection analyses showed that selective sweeps, and in particular those that are recent and ongoing (*XP-nSL*), were pervasive across the genomes of urban great tits, supporting our inference of a highly redundant polygenic adaptation to urban environments. The underpinning genomic changes associated with urban adaptation were largely population specific, likely the result of differential selective pressures, adaptation through complex multivariate phenotypic changes and/or selection on standing genetic variation[49,50]. Nonetheless, we still detected shared selective sweeps across urban populations, suggesting the presence of common genomic responses to the urban selective pressures, either through repeated selective sweeps or sharing of adaptive genetic variants.

**Evolutionary drivers of genetic differentiation and signatures of selection**. The largely population-specific landscapes of differentiation are in line with observations in other young divergences, and have been previously explained by the weaker effect of linked selection at early stages of the divergence process[47]. However, to exclude a driving role of non-adaptive processes, we corroborated that the detected signatures of selection were indeed caused by divergent selection and not by genetic drift or linked selection in low-recombination regions[47]. To achieve this, we evaluated the correlation between signatures of selection (*XP-nSL* and *Rsb*) and differentiation ($F_{ST}$) with two proxies of genomic recombination rate, i.e., LD and intronic GC-content (both in

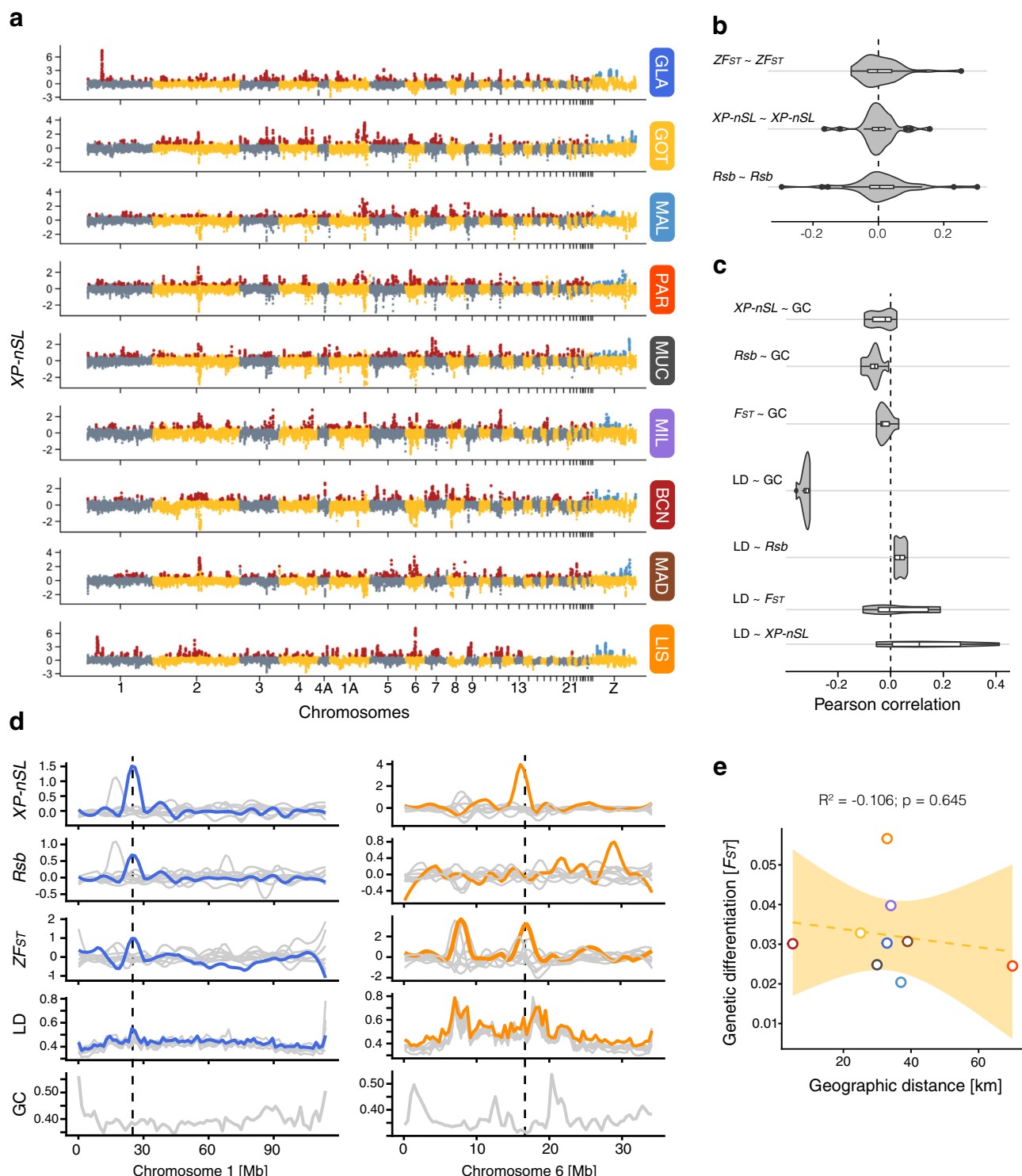

200 kb non-sliding windows). LD is correlated with population-specific recombination rates and the strength of selection, while intronic GC-content is a surrogate for the long-term recombination rate variation across the great tit genome, i.e., a higher GC-content will indicate a higher recombination rate[51,52]. To confirm that both proxies provide similar pictures of the recombination landscape, we tested their correlation across all population pairs. Indeed, LD and GC-content, were (negatively) correlated in all populations (Fig. 4c), suggesting that the general effect of recombination on the genomic diversity landscape is broadly consistent.

Under a scenario of linked selection in low-recombination regions, we would expect strong and consistent positive correlations between GC-content or LD with our estimates of genetic differentiation and selection, i.e., $F_{ST}$, XP-nSL and Rsb[53–55]. Yet, this was not the case, (Fig. 4c), which indicates that processes other than linked selection are the main driver for population differentiation in our study. Indeed, the overall patterns are more in line with divergent selection, as e.g., the population-specific selection peak on chromosome 1 in Glasgow is not associated with low-recombination regions (low GC-

**Fig. 4 Signatures of selection in urban populations. a** Manhattan plots showing the distribution of cross-population $nSL$ scores ($XP\text{-}nSL$) for each urban–rural population pair in 200 kb sliding windows with a 50 kb step size. Positive scores depict selection in urban populations. Significant autosomal outlier windows are highlighted in red and outlier windows on the Z-chromosome in blue (see "Methods"). Autosomal and sex-chromosome outlier windows were detected separately. **b** Violin plots showing the distribution of correlation coefficients (Pearson's $r$) for the different inter-population summary statistics for all pairwise comparisons. Note that all distributions overlap zero and are not consistent across pairs. The shaded grey areas of the violin plot denote the kernel density estimation of the data distribution. The bottom and top edges of the white boxplot denote the upper and lower interquartile range (25th and 75th) with the vertical black line showing the median, and the thin horizontal line representing the rest of the distribution. Black dots represent values outside the distribution. **c** Violin plots for the distribution of correlation coefficients between inter-population summary statistics with proxies for recombination rate (within population LD, GC-content). The boxplot shows the interquartile range (white area), median (vertical black line) and the remaining distribution (horizontal black line). **d** Example signatures of divergence and proxies for recombination along chromosome 1 and 6, to highlight the heterogeneity of divergence across populations. Window-based measures were loess-smoothed across each chromosome. Populations with selective sweeps on the respective chromosomes are highlighted, with Glasgow (blue) on chromosome 1 and Lisbon (orange) on chromosome 6. All other populations are in grey in the background. Note that the highest sweep window, marked by the dotted line, corresponds to a likely low-recombination region on chromosome 6, but a normal recombination region on chromosome 1. **e** Non-significant correlation of the geographic distance between urban and rural populations from the same locality and their genetic differentiation. (linear model: $F_{1,7} = 0.23$, $P = 0.645$). Shaded area denotes the 95% CI. BCN Barcelona, GLA Glasgow, GOT Gothenburg, LIS Lisbon, MAD Madrid, MAL Malmö, MIL Milan, MUC Munich, PAR Paris. Source data are provided as a Source data file.

content) or a pronounced peak in LD (Fig. 4d). However, it is important to note that these correlations are positive in some populations, see e.g., $XP\text{-}nSL$ and GC-content (Fig. 4c), and that some population-specific peaks are located on low-recombination regions, and likely caused by linked selection or other non-adaptive processes (Fig. 4d). While the contribution of non-adaptive processes likely differed across populations, we did not detect a general signature of linked selection as a major driver of divergence between urban and rural populations.

Divergent selection has been suggested to be the main driver in the early stages of differentiation in other avian systems[47,56], and it is not surprising that a similar incipient pattern is observed between urban–rural populations. Although other non-adaptive processes, such as stochasticity via population-specific drift, could also generate variable genomic landscapes in an urban context[57], that scenario would be characterised by a decreased gene flow and increased genetic differentiation with geographical distance between adjacent urban and rural populations. However, genetic differentiation between paired urban and rural populations did not significantly increase with geographic distance ($R^2 = -0.11$, $P = 0.645$; Fig. 4e), further excluding drift as the main process underneath the variability in the genomic and the genetic differentiation landscape between urban and rural populations. Nonetheless, genetic drift potentially led to increased genetic differentiation in some urban populations, i.e., Glasgow and Lisbon, which both showed reduced heterozygosity.

Hence, we conclude that the mosaic of genomic signatures of differentiation between urban and rural populations and selective sweeps in the studied populations are most likely driven by genomic responses to selection on standing genetic variation, with potentially minor contributions of linked selection in low-recombination regions or other idiosyncratic non-adaptive processes.

**Genes and pathways associated with adaptation to urbanisation.** While responses to selection at the haplotype level were highly variable across urban populations, selective sweeps might affect the same genes or different genes with similar functional impacts (e.g., same pathway). Such consistent genomic responses on the gene or functional level could explain the similarity in phenotypic responses to environmental variation despite a large variation on the haplotype level[14,15,58]. Following this rationale, we identified genes associated with signatures of selective sweeps in each urban population, but without requiring outlier windows to overlap with each other. Between 976 and 1366 genes were associated with signatures of ongoing or recent selective sweeps

($XP\text{-}nSL$), and 362–923 with completed selective sweeps ($Rsb$). Of these genes, 64–207, and 5–76, were detected in at least two urban populations (based on $XP\text{-}nSL$ or $Rsb$, respectively; Supplementary Figs. 15 and 16). The higher number of genes associated with recent and ongoing selective sweeps compared to complete selective sweeps is in line with the colonisation pattern of urban habitats, under ongoing or recent gene flow.

Although it is statistically unlikely to detect selective sweeps in the same gene in more than three urban populations by chance alone (permutation test; $\chi^2_1 = 77.95$, $P < 2.2 \times 10^{-16}$), we conservatively focused our functional interpretation only on those genes that were associated with signatures of selection ($XP\text{-}nSL$ or $Rsb$) in more than half of all urban populations ($\geq 5$ urban populations), as these likely play more important roles in urban adaptation. In total, we detected 42 genes associated with recent or ongoing selective sweeps in at least five populations ($XP\text{-}nSL$: max. six populations), and 15 genes associated with older completed selective sweeps ($Rsb$: max. seven populations; Fig. 5a and Supplementary Table 4), with none of the genes detected by both selection statistics. It is noteworthy that we did not detect an increase in the number of shared genes under selection in relation to geographical (Mantel's $r = 0.27$, $P = 0.080$) or genetic distance (Mantel's $r = 0.05$, $P = 0.480$), which is in accordance with the results regarding shared outlier windows and further supports that many selective sweeps likely occurred independently in multiple urban populations. In addition, 12 of the 57 shared genes under selection were also associated with urbanisation in the LFMM analysis (Supplementary Table 4). Interestingly, two of these, $GMDS$ and $SLC6A15$, were not only associated with complete selective sweeps ($Rsb$ scores) in seven and six populations, respectively, but also previously shown to be differentially expressed in blood and/or liver between urban and rural birds from Malmö[59]. Moreover, following Malmö as an example, one gene found to be differentially expressed in Watson et al.[59], $VPS13A$, was highly differentiated ($ZF_{ST}$) and associated with selective sweeps in that particular population, but importantly, also under selection in Madrid ($Rsb$) and Lisbon ($XP\text{-}nSL$) and in general, associated with urbanisation (LFMM analysis). This exemplifies that some genes show signatures of urban adaptation on a continental scale, making them strong candidates for adaptation in these urban centres.

It is noteworthy that despite finding signatures of selection overlapping the same gene in five or more urban populations (e.g., $GMDS$, $HTR7$ or $CDH18$), the exact locations of these signatures were not consistent (Fig. 5b, c). Under a shared selective sweep or adaptive introgression scenario, we would have

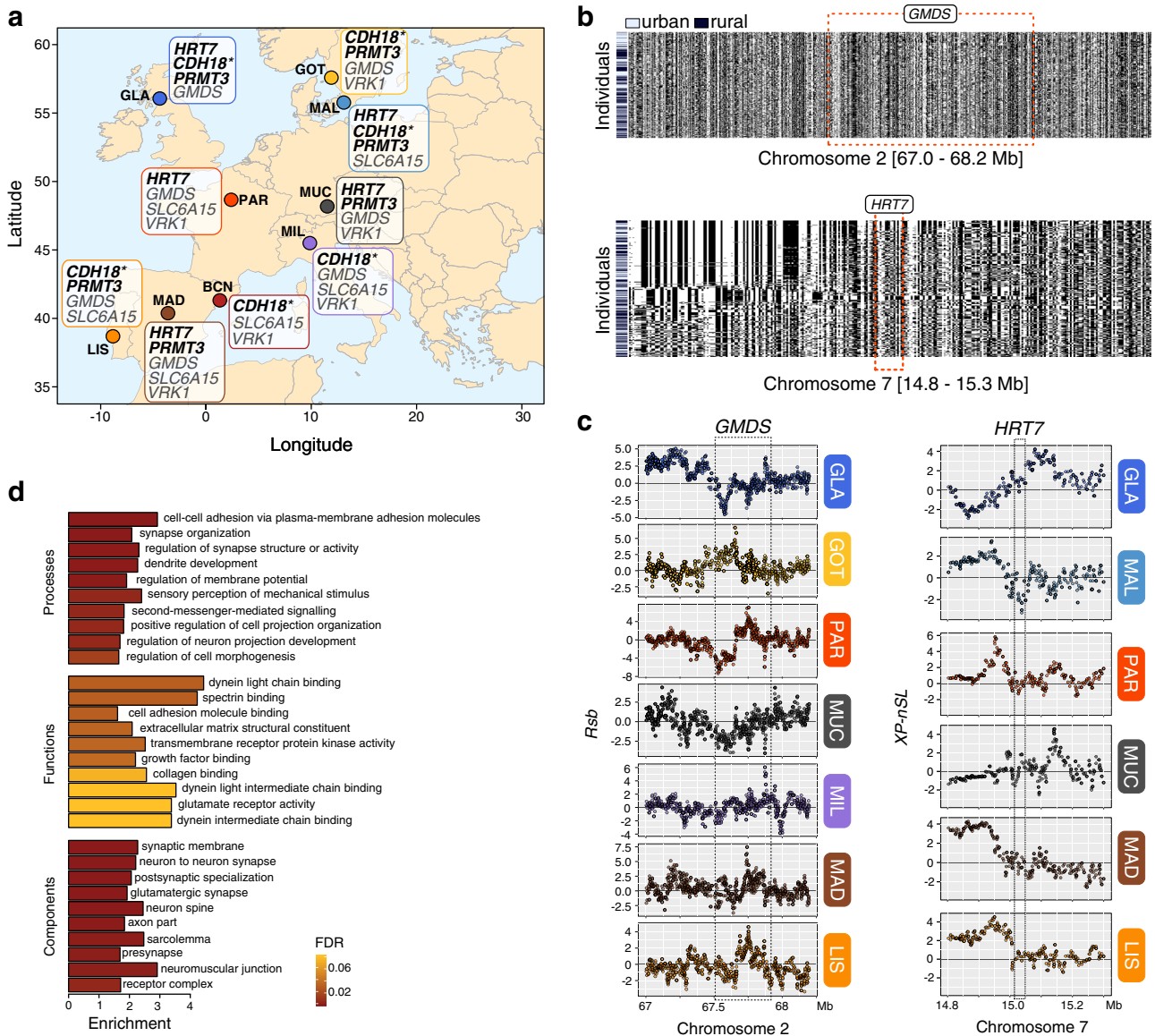

**Fig. 5 Candidate genes and pathways underlying urban adaptation. a** Map showing the spatial distribution of the top shared candidate genes putatively under selection in urban populations across Europe, detected in at least six urban populations. *CDH18*, marked with an asterisk, was also detected in the GEA analysis. Genes in bold font were associated with ongoing selection (*XP-nSL*), and those with grey font with older completed sweeps (*Rsb*). The map was plotted in R using ggplot with the maps package. **b** Haplotype strips showing haplotype structure around two representative candidate genes (*GMDS* and *HRT7*) on chromosome 2 and 7, respectively. Individuals (rows) are ordered by haplotype similarity and colour-coded by habitat of origin. Note that individuals from the same habitat do not share the same haplotype. **c** Distribution of SNP-based *Rsb* scores for each of the six populations, in which *GMDS* is associated with significant outlier windows, and *XP-nSL* scores around *HRT7*. Red shades depict SNPs with the strongest positive selection scores. Gene boundaries are shown by red dotted lines and grey background box. Note that not always the same regions (upstream, downstream or genic) are under selection in urban populations (positive scores) around the candidate genes. **d** Enrichment of associated genes in significantly enriched gene ontology (GO) groups by GO category in the LFMM and BayPass analysis (see Supplementary Table 4 for a detailed list). The colour of the bars represents the false-discovery rate (FDR) from the gene ontology overrepresentation analyses. BCN Barcelona, GLA Glasgow, GOT Gothenburg, LIS Lisbon, MAD Madrid, MAL Malmö, MIL Milan, MUC Munich, PAR Paris.

expected the sharing of urban haplotypes between populations. However, haplotype plots do not show any noticeable clustering across urban individuals (e.g., *GMDS* or *HTR7*, Fig. 5b), suggesting that different adaptive haplotypes swept to high frequency in urban populations across Europe, which is in line with soft selective sweeps from standing genetic variation[60]. Furthermore, we find that the strongest selective sweep signatures are, in some cases, not within the genes but up- or downstream and without a consistent pattern among populations. For example, signatures of selection around *HTR7* were downstream

in Glasgow and Munich, but upstream in Malmö, Paris, Madrid and Lisbon (Fig. 5c). This might suggest that regulatory changes rather than protein-coding ones are associated with urban adaptation and the nature of regulatory variation likely differs across populations. However, high-resolution genomic studies based on whole-genome resequencing coupled with multi-tissue functional genomic analyses are needed to resolve this question, which is beyond the scope of this study.

Many of the genes associated with urbanisation have previously been linked to behavioural divergence, and cognitive and learning

functions, suggesting adaptive phenotypic shifts related to behaviour. In particular, *HTR7* (Chr 7) codes for a receptor of the neurotransmitter serotonin, a pathway consistently identified as target of changes linked to urbanisation. Indeed, two previous studies in birds, one on European blackbirds using a candidate gene approach and another on burrowing owls (*Athene cunicularia*), using full-genome resequencing, also described changes in this pathway as the main difference between urban and rural populations[3,22]. Both species also show consistent urban–rural differences in two behaviours associated with serotonin: down-regulation of the stress axis and altered nocturnal behaviour under artificial light at night, respectively[9,61,62]. The same gene, *HRT7*, has also been directly linked to behavioural differences between migratory and resident rainbow trout (*Onchorynchus mykiss*)[63], supporting its potential role in behavioural changes in urban great tits. The *CDH18* gene (Chr 2), is part of a superfamily of membrane proteins involved in synaptic adhesion and revealed as a candidate gene in phonological alterations in humans[64]. The *PTPRD* and *VPS13A* genes (both in Chr Z), are suggested to be involved in hippocampal neural development[65] and were previously linked to bird navigation, flight performance and migratory behaviour[66–68]. Moreover, two other genes under selection and associated with urbanisation are involved in the regulation of cognitive processes and behavioural disorders in vertebrates (*DLG2* (ref. [69]) and *NRXN3* (ref. [70]), Chr 1 and 5, respectively), and were also previously linked to urban divergence in burrowing owls[22].

Thus, our selection analyses suggest that natural selection repeatedly acts on behavioural traits and sensory and cognitive performance, all previously shown to be among the most widespread differences between urban and rural wildlife populations[71,72]. These findings were furthermore supported by gene ontology (GO) analysis of the 2758 urbanisation-associated SNPs (LFMM analysis), linked to 984 genes (1501 SNPs in genic regions). Accordingly, most of the GO terms were related to neural functioning and development (e.g., GO:0016358, $FDR = 4.30 \times 10^{-6}$), cell adhesion (e.g., GO:0098742, $FDR = 3.43 \times 10^{-6}$) and sensory perception (e.g., GO:0050954, $FDR = 3.42 \times 10^{-3}$; Fig. 5d and Supplementary Table 5). These GO terms were mainly clustered into two interacting networks, one related to sensory recognition and the other to neural development and cell adhesion (Supplementary Fig. 17). These findings reinforce the previous idea on cognitive and behavioural changes as key responses to urbanisation, as song structure and escape or distress behaviour have been previously shown to differ between urban and rural great tit populations across Europe[7,35,73]. Nonetheless, whether this is the result of a genetic response to selection or phenotypic plasticity was to a large extent still unknown. Our present study suggests a strong genetic component for these putatively urban adaptative phenotypes in great tits across Europe. Detailed functional genomic and phenotypic analyses are now needed to understand the role of these genes and pathways in the adaptive divergence of urban and rural great tits and other songbirds.

Our study demonstrates clear genomic signals of local adaptation to urban habitats in a common songbird on a continent-wide scale across Europe. We found that a combination of polygenic allele frequency shifts and spatially varying selective sweeps are associated with adaptation to urban environments. Our results strongly suggest that a few genes, which have known neural developmental and behavioural functions, experienced selective sweeps in urban populations. This suggests a strong consistency in the functional processes associated with urbanisation, despite the fact that the underlying haplotypes are often not shared. Thus, our study exemplifies evolutionary adaptation to urban environments on a European scale, and highlights behavioural and neurosensory adjustments as important phenotypic adaptations in urban habitats.

## Methods

**Sample collection and DNA extraction**. During the years 2013–2015, 20 or more individual great tits were sampled at paired urban–rural sites from nine European cities (Fig. 1a and Supplementary Table 1). We sampled a total of 192 individuals (aged >1 year old) with 10–16 individuals per site (Supplementary Table 1). Sexes were balanced between pairs (urban–rural) in the dataset (GLMM; sex: $\chi^2_1 = 1.33$, $P = 0.280$). Each of the paired sampling sites (urban or rural, hereinafter populations) was sampled within the same season. Barcelona and Munich were sampled during winter, however, in both cases only known birds (recaptures) were included in the study, thus, all birds can be considered resident. All urban populations were located within the city boundaries, the areas are characterised with significant proportion of human-built structures, such as houses and roads with managed parks as the only green space (Supplementary Fig. 1). Rural populations were chosen to contrast the urban locations regarding degree of urbanisation and were always natural/semi-natural forests, and contained only a few isolated houses. All urban and rural populations were separated by a distance above the mean adult and natal dispersal distance of this species (i.e., see Supplementary Table 1)[74].

Blood samples (~25 µl) were obtained either from the jugular or brachial vein and stored at 4 °C in ethanol or SET buffer and subsequently frozen at −20 °C. In each case, procedures were identical for the paired urban and rural populations. All procedures employed during field work were approved by national Ethical Committees and authorisations to collect samples were delivered by the Environment Department of the Generalitat de Catalunya (permit no. AECC/SF/ 0438), the Scottish Natural Heritage (permit no. 52463) and UK Home Office (license no. 70/7899), the Malmö-Lund animal Ethical Committee (permit no. M454 12:1), the CEMPA and Portuguese Ministry of Environment (permit nos. 40/ 2014 and 164/2014), the Ministry of the Environment, Housing and Territorial Planning of Madrid (permit nos. 10/103329.9/14, 10/169940.9/13, 10/045383.9/14, 10/127641.9/14 and 10/055393.9/14), the Institute for Environmental Protection and Research (ISPRA, license nos.15510 and 15944) and the Lombardy Region (permit no. 3462), the Tierschutzgesetz (TierSchG, German animal protection law) and the Regierung von Oberbayern (permit nos. 55.2-1-54-2532.2-7-07 and 55.2-1-54-2532-140-11) and the Minister of Higher Education, Research and Innovation (Ethics Committee for Animal Experimentation license no. 005 and permit no. APAFIS#19941-2019032516275025), the Prefect of Paris and the Prefect of Seine et Marne (permit nos. DRIEE-2012-31 and DRIEE-2012-32) and the CRBPO (National Museum of Natural History, permit no. 537 and licence no. 1454). DNA was extracted from ~5 µl samples of red blood cells in 195 µl of phosphate-buffered saline, using Macherey-Nagel NucleoSpin Blood Kits (Bethlehem, PA, USA), and following the manufacturer's instructions or manual salt extraction (ammonium acetate). The quantity and purity of the extracted genomic DNA was high as measured using a Nanodrop 2000 Spectrophotometer (Thermo Fisher Scientific) and Qubit 2.0 Fluorometer (Thermo Fisher Scientific).

**Urbanisation score**. To quantify the degree of urbanisation at each site, we used the UrbanizationScore image analysis software[75], based on aerial images from Google Maps (Google Maps 2017), and following the methods described in different studies assessing the effect of urbanisation on wild bird populations[76]. Briefly, each sampling site was represented by a 1 × 1 km rectangular area around the capture locations. We opted to use this spatial resolution around the sampling sites to approximately cover individuals' territory[32], but also to ensure that we considered the heterogeneity within each selected site. The content in each rectangle was evaluated dividing the image in 100 × 100 m cells and considering three land-cover characteristics in each: proportion of buildings, vegetation (including cultivated fields) and paved surfaces (Supplementary Fig. 1). The different land-cover measures obtained per site were used in a PCA to estimate an urbanisation score variable ($PC_{urb}$) for each of the urban or rural populations per locality, see Table S1. The $PC_{urb}$ values were transformed to obtain negative values in the less urbanised and positive values in the more urbanised sites. We used the average of the urbanisation estimates if birds were captured in more than one location within each site (>2 km apart, mean ± SD: 931.22 ± 1005.26 m). All quantifications were done in triplicates by the same person (P.S.) and the estimates were highly repeatable (intra class correlation coefficient, ICC = 0.993, 95% confidence interval (CI) = 0.997–0.987, $P < 0.001$).

**SNP genotyping**. All 192 individuals were successfully genotyped using a custom made Affymetrix© great tit 650 K SNP chip at Edinburgh Genomics (Edinburgh, United Kingdom). The Affymetrix SNP chip was developed based on whole-genome sequencing data from great tits sampled across multiple European populations, largely corresponding to our sample sites[38]. Thus, this SNP chip allowed us to genotype a large number of individuals on a genome-wide basis at high SNP density without a strong ascertainment bias. SNP calling was done following the "Best Practices Workflow" in the software Axiom Analysis Suite

1.1.0.616 (Affymetrix©) and all the individuals passed the default quality control steps provided by the manufacturer (dish quality control values >0.95) and previous studies using the same SNP chip[38,77]. A total of 544,610 SNPs were then exported to a variant-calling format (VCF) and Plink v. 1.9 (ref. [78]), and further filtered and assigned to chromosomes using the *P. major* reference genome build 1.1 (annotation release ID 101; "GCA_001522545.2"). A total of 155 SNPs were not found in the used assembly and 26,852 SNPs were not in chromosomic regions; thus, these SNPs were removed from further analysis leaving a total of 517,603 SNPs.

**Genetic diversity and population structure**. We calculated the genome-wide genetic diversity as expected heterozygosity ($H_e$) for each population using Plink v. 1.9 (ref. [78]), and tested if genetic diversity significantly differed between urban–rural populations from the same location using $t$ tests in R, and overall across urban–rural pairings using a Wilcoxon rank-sum test in R v.3.6.1 (ref. [79]). Furthermore, we estimated pairwise $F_{ST}$ between all population pairs (urban–rural per locality), using VCFtools v. 0.1.15 (ref. [80]). Mean average $F_{ST}$ was computed across all comparisons after setting negative values to zero.

For analyses of population structure, we pruned the SNP dataset based on LD in Plink v.1.9 (ref. [78]), using a variance inflation factor threshold of 2 ("-indep 50 5 2"), retaining 358,149 SNPs. Using this LD-pruned dataset, we performed a PCA using Plink. We performed two different PCAs, (i) only with autosomes and excluding all small linkage groups and the Z-chromosome, and (ii) including the Z-chromosome but excluding all small linkage groups. Subsequently, we performed additional dimensionality reduction for each PCA dataset using the first 20 principal components, using the UMAP v.0.2.7.0 R-package with default settings[81]. We also compared the results to the UMAP based on the first ten PCs. Genetic ancestry analysis was performed using the software package fastStructure v.1.0 (ref. [82]), with K ranging from 2 to 9 and cross-validation. In addition, we inferred a population tree based on allele frequency co-variances using Treemix v.1.3 (ref. [83]), with blocks of 100 SNPs. We fitted up to five migration edges and determined the optimal number of migration edges by (i) comparing the likelihoods of trees with different number of migration edges, (ii) estimating the total variance explained by each tree and (iii) by comparing the significance of the fitted migration edges.

To infer migration rates across our entire sampling area, we used EEMS[84] to estimate effective migration surfaces across all our samples based on the complete LD-pruned SNP dataset, with the following settings: nDemes = 1000, numMCMCIter = 2,000,000, numBurnIter = 1,000,000 and numThinIter = 9999. The results were plotted using the R-scripts provided with the EEMS software packages.

**Environment-associated SNPs**. We used two different approaches to identify SNPs associated with the degree of urbanisation, based on the "urbanisation score". First, we used a univariate latent-factor linear mixed model implemented in LFMM v.1.5 for examining allele frequency–environment associations[85]. Based on the number of ancestry clusters (K) inferred with fastStructure v.1.0, we ran LFMM with two and four latent factors, respectively. Each model was run five times for 10,000 iterations with a 5000-iteration burn-in. We calculated the median $z$-score for each locus across all ten runs, and selected SNPs with a FDR < 1% to be associated with urbanisation. The results with two or four latent factors were highly concordant and the same candidate loci were recovered; thus, we only used the results obtained with four latent factors for further analyses. We also assessed the distribution of $P$ values to check for the impact of confounding factors (Supplementary Fig. 5a). In addition, to assess if associations are putatively false positives, we performed a permutation analysis. We randomised habitat-assignments 20 times and performed LFMM association analyses on each randomised dataset. Following Fuller et al.[86] we determined a significance threshold as the 95th percentile of the $Z$-score distribution and identified SNPs from the initial LFMM analyses above this threshold as significant. Using the significance cut-off based on randomisation (>99th percentile), we detected a far higher number of associated SNPs (4358 SNPs; Supplementary Fig. 5b) than using a FDR < 1% (2758 SNPs; Fig. 3a); therefore, and to avoid a larger number of false positives, we opted for a more conservative approach and focus on the FDR results in the analysis (see "Results and discussion").

Second, we analysed associations with urbanisation using the auxiliary covariate model implemented in BayPass v.2.1 (ref. [87]). We estimated the allele frequency–environment association for each SNP with the urbanisation score for each population accounting for population structure, using a covariance matrix. We estimated the covariance matrix using the LD-pruned SNP dataset in the core model using default parameters: 20 pilot runs of 1000 iterations, a run length of 50,000 iterations, sampling every 25th iteration, and a burn-in of 5000 iterations. The resulting covariance matrix was used as input for five replicated runs of the auxiliary covariate model using the above settings. The strength of association is given in the test by estimated Bayes factor (measured in deciban; dB). We calculated the median Bayes factor across all five replicated runs and considered all SNPs with a deciban unit (dB) > 20 as urbanisation associated. This is the strictest criterion and is considered as "decisive evidence" for the association[87]. Using a resampling-without-replacement-based permutation approach and a chi-square

test, we determined if the overlap between the LFMM and BayPass candidate SNPs is higher than expected by chance.

**Patterns of genetic differentiation ($F_{ST}$)**. To identify genomic regions distinguishing adjacent urban and rural great tits, we estimated the genetic differentiation (Weir and Cockerham's $F_{ST}$[88]) for each urban and rural pair for each SNP, using VCFtools v. 0.1.15 (ref. [80]). We subsequently summarised and plotted $F_{ST}$ values in 200 kb sliding windows with 50 kb steps, using the WindowScanR v.0.1 R-package (https://github.com/tavareshugo/WindowScanR), and standardised window-based $F_{ST}$ estimates using a $z$-transformation ($ZF_{ST}$). Standardisation was performed separately for autosomes and the Z-chromosome.

To determine if the extent of genetic differentiation between all possible population pairs was significantly different from zero, we estimated $P$ values using a permutation analysis in Genodive v.3 (ref. [89]). For computational reasons, the analysis was performed on a thinned SNP dataset (21,062 SNPs), for which we randomly selected one SNP every ~50 kb.

**Haplotype-based selection analyses**. To identify genomic regions showing signs of selective sweeps in urban–rural population pairs, we used two haplotype-based selection scans. First, we scanned the genome for regions showing differences in extended haplotype homozygosity (EHHS) between urban and rural populations (see $Rsb$ score below) to identify genomic regions showing signs of older and completed selective sweeps. We used fastPHASE v.1.4 (ref. [90]) to reconstruct haplotypes and impute missing data independently for each chromosome using the default parameters, except that each individual was classified by its population ("-u" option). We used ten random starts of the EM algorithm ("-T" option) and 100 haplotypes ("-H" option). The fastPHASE output files were analysed using rehh v.2.0 (ref. [91]) to calculate $Rsb$ statistics per focal SNP. The $Rsb$ score is the standardised ratio of integrated EHHS (iES, which is a site-specific EHHS) between two populations[48,91]. This statistic measures the extent of haplotype homozygosity between two populations and follows the rationale that if a SNP is under selection in one population compared to the other, the region around this locus will show an unusually high level of haplotype homozygosity compared to the neutral distribution. In accordance with Gautier and colleagues[91], we considered significant SNPs putatively under selection in urban populations based on a threshold of $Rsb \geq 4$. To identify genomic regions consistently showing strong signals of selection, we summarised $Rsb$ scores in 200 kb sliding windows (50 kb steps) and selected windows under selection as those with (i) an average $Rsb$ scores above the 95th percentile of the genome-wide distribution (separately for autosomes and the Z-chromosome) and (ii) a proportion of urban outlier SNPs ($Rsb > 4$) above the 95th percentile of the genome-wide distribution (separately for autosomes and the Z-chromosome). The proportion of outlier SNPs per window was the number of SNPs with $Rsb$ values >4 compared to the total number of SNPs per window[92]. Because recombination rates can be assumed to be conserved between closely related urban and rural populations, the cross-population comparative nature of the $Rsb$ statistic provides an internal control that cancels out the effect of heterogeneous recombination across the genome[48].

Second, we used a cross-population $nSL$[93] statistic (XP-$nSL$) that tests for signatures of ongoing or recently completed selective sweeps and is implemented in selscan v.1.3.0 (ref. [46]). XP-$nSL$ contrasts $nSL$, the number of segregating sites by length, between two populations and thus tests for differential local adaptation. Analogous to $Rsb$, we conservatively detected genomic regions by a 200 kb sliding windows as those with (i) average XP-$nSL$ scores above the 95th percentile of genome-wide distribution (separately for autosomes and the Z-chromosome) in each population and (ii) proportions of outlier SNPs (XP-$nSL > 2$) above the 95th percentile. We also plotted haplotype patterns around candidate genes using the haplostrips software v.1.3 (ref. [94]).

To determine larger genomic regions associated with signals of selection, we merged adjacent outlier windows for each selection statistic (XP-$nSL$ and $Rsb$) into outlier regions, if windows were not >200 kb apart, using csaw v.3.12 (ref. [95]).

To determine the consistency of selection across urban centres, we implemented a resampling approach to assess the likelihood of genes showing signs of selection in two, three, four or more populations. We resampled with replacement $n$ genes ($n$ = number of genes with signatures of selection in each urban population) for each population from the list of all SNP-linked genes using the "resample" function in R, assessed the amount of overlap between populations (from two to eight populations) and repeated the sampling 100,000 times for each comparison. We then calculated the mean and 95% CI for each comparison and compared the number of observed shared candidate genes to the expected number of candidate genes. The expected number of genes showing signs of selection in three or more populations was zero, thus we focused on genes showing signs of selection ($Rsb$ and XP-$nSL$) in three or more populations.

**Decomposing habitat and locality effects for urbanisation-associated SNPs**. To determine the explanatory power of urbanisation-associated SNPs and consistency in allele frequency changes across populations, we used linear models (i.e. "Y ~ habitat × locality × habitat × locality") to quantify the effects of (i) "habitat" (i.e., consistent change in allele frequency across urban–rural pairs), (ii) "locality" (i.e., effect of city-of-origin on absolute allele frequency) and (iii) "habitat ×

locality" interaction (i.e., inconsistent change in allele frequency across urban–rural pairs)[44,96]. We performed this analysis for all individual SNPs (coded as 0, 1 or 2) and based on PC scores from the first three PC axes for all LFMM candidate SNPs ($PC_{LFMM}$), and those overlapping between the LFMM and BayPass analyses ("core urbanisation SNPs", $PC_{GEA}$, see above). The PCA was estimated for each SNP dataset using SNPrelate v.3.12 (ref. [97]). We used the "EtaSq" function implemented in BaylorEdPsych v.0.5 (ref. [98]) to extract the effect sizes (partial $\eta^2$) for the model terms in each linear model.

We further estimated the directionality of allele frequency changes across populations by counting in how many urban populations the same allele was the minor allele for all significant LFMM candidate SNPs and the "core urbanisation SNPs". We then estimated for each SNP dataset the proportion of SNPs that showed concordant minor alleles in five, six, seven, eight or nine urban populations.

**Patterns of linkage disequilibrium and intronic GC-content across the genome**. To estimate the impact of variation in recombination rate and linked selection in low-recombination regions on patterns of divergence (i.e., regions showing signs of selective sweeps), we estimated the correlations of LD and intronic GC-content in 200 kb windows with measures of selection/differentiation ($Rsb$, $XP\text{-}nSL$ and $ZF_{ST}$).

First, we estimated patterns of LD ($r^2$) across the genome using Plink 1.9 (ref. [78]) for pairs of SNPs located up to 200 kb apart for each urban population (–r2–ld-window-r2 0–ld-window-kb 200,000). LD estimates were averaged in 200 non-sliding windows using the WindowScanR v.0.1 R-package.

Second, we estimated the GC-content for all introns across the great tit (*P. major*) reference genome build 1.1 (annotation release ID 101; "GCA_001522545.2"), as low intronic GC-content is a good proxy for long-term reduced recombination in that region[52]. Intron coordinates were extracted from the great tit reference genome using the plyranges v.3.12 (ref. [99]) and GenomicRanges v.3.12 (ref. [100]) R-packages, and the GC-content for each intron inferred using the "nuc" function in BEDtools v.2.28 (ref. [101]). Intronic GC-content values were further averaged across 200 kb non-sliding windows using WindowScanR.

Lastly, we estimated the genome-wide Pearson correlations between LD, GC-content and all estimators of selection ($Rsb$, $XP\text{-}nSL$ and $ZF_{ST}$) across all 200 kb windows for each population, using the "cor.test" function in R.

**Functional characterisation of candidate SNP**. We obtained the gene annotations for all candidate SNPs from the great tit (*P. major*) reference genome build 1.1 (annotation release ID 101; "GCA_001522545.2"). We used all genes containing SNPs associated with urbanisation (LFMM and BayPass, $n = 1501$ SNPs within genes). To analyse the enrichment of functional classes, we identified over-represented GOs (biological processes, molecular functions and cellular components), using the WebGestalt software tool[102]. The gene background was set using annotated great tit genes (annotation release ID 101) containing SNPs from the SNP chip and with *Homo sapiens* orthologues. *H. sapiens* genes were used as a reference set as human genes are better annotated with GO terms than those of any avian system (e.g., *Gallus gallus*)[16,39]. We focused on non-redundant GO terms to account for correlations across the GO graph topology and GO terms as implemented in WebGestalt[102]. An FDR < 0.05 was used as a threshold for significantly enriched GO terms. Furthermore, we searched the public record for functions of individual candidate genes. We also used GOrilla[103] to visualise the connections of GO terms associated with LFMM candidate genes and Cytoscape v.3.6.1 (ref. [104]) to visualise the GO network and identify all enriched GO terms (biological processes, $P < 0.001$), including redundant terms.

Candidate genes associated with signatures of selection were those that overlapped with significant $XP\text{-}nSL$ or $Rsb$ outlier windows. Overlaps between outlier windows and annotated genes were assessed using the plyranges R-package v.3.12.

**Reporting summary**. Further information on research design is available in the Nature Research Reporting Summary linked to this article.

## Data availability

The genotyping data that support the findings of this study is available in variant call format (VCF) via the European Variation Archive (EVA) with the accession number PRJEB44069. Source data are provided with this paper.

## Code availability

No custom code or mathematical algorithm was developed for this project or is considered crucial to the conclusions. All relevant software and R-functions that were used are referred to in the "Methods" section.

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

## Acknowledgements

We are truly thankful to J. Pérez-Tris, J. I. Aguirre de Miguel, M. Morganti, F. Spina, ringers from La Herreria and Lago di Pusiano, V. Encarnação, A. Mouchet and Pardal family. We also thank V.N. Laine, A. Herrera-Dueñas and A.C. Mateman for their help during data extraction and formatting and laboratory work, and M. Bosse for comments on an early version of the manuscript. Funding was provided by the Swedish Research Council C0361301 and Marie Curie Career Integration Grant FP7-CIG ID:322217 (to C.I.), Ministry of Economics and Competiveness (CGL-2016-79568-C3-3-P to J.C.S).

## Author contributions

C.I. and P.S. conceived the study. P.S. collected and coordinated the sample collection with additions from C.B., N.D., D.M.D., B.H., J.C.S. and Ph.S. The data analysis was carried out by A.J. and P.S. with help from M.L., D.A. and B.H.; M.E.V., C.I., A.J. and P.S. drafted the initial manuscript with input from all the other authors.

## Funding

## Competing interests

The authors declare no competing interests.
