## [Peer Review File · Nature Communications]

REVIEWER COMMENTS

Reviewer #1 (Remarks to the Author):

In this paper, Salmon et al. mined the genomes of nine urban-rural population pairs of Great Tits for genomic regions underlying adaptation to urban environments. The paper present comprehensive analyses that reveal that many independent loci appear to underlie urban adaptation. Nevertheless, the authors also identify genomic regions in which selective sweeps occurred in several urban populations. The identified genomic regions are predominantly related to neutral function and development.

I think this is an interesting paper that could make for a nice contribution to Nature Communications. Nevertheless, I have a couple of comments that are in part related to presentation and in part to analysis and interpretation. My main comment is probably with regard to the analysis of population structure. These do in my opinion not provide a clear picture of how populations are related; notably there are a result or two that need explaining. Please see detailed comments.

Detailed comments

25-26: Written as is, it is unclear not entirely clear what 'localized' means and why this hampers replicability. Please rephrase.

29: Are these sweeps independent or the same, and what is the evidence?

29-32: It is not entirely clear how the association with the two specific functions points to adaptation through the same pathways. Please rephrase.

44: Evolutionary signals of what? What are 'evolutionary signals'? What are 'signals of adaptation' (45)?

49/50: Written as is, this kind of takes for granted that there is convergent evolution. I would suggest introducing this differently. In my mind, the first question is whether there is parallel/convergent evolution at all and what its origins are.

52: Please delete 'strong'

54: Here the authors write about 'parallel evolutionary adaptation' (versus convergent evolution before). It would in my opinion be important to stick with one and clearly define what the definition entails, as there are different definitions of each.

I have the feeling that the way parallel/convergent evolution is presented here mingles two things: The prevalence of the latter per se, and parallel responses to selection as a tool to confirm findings in one population with the same findings in other populations. It is unclear to me, which of these the authors

address in first line. I would suggest adapting the introduction accordingly.

69: What was the goal of these two approaches?

80: F_{st} values of up to 5% at such small geographic distance do not represent low differentiation at such a local scale. If significantly different from zero, even smaller values would seem rather substantial for passerine birds that usually show high dispersal rates. It would therefore seem important to test whether differentiation is different from zero. Moreover, how does local differentiation compare to differentiation at the continent level? When I look at the values, it would seem that differentiation at the local scale is comparable to differentiation between populations. The question is in this case indeed, whether these values are significantly different from zero, if there is not the least pattern of isolation by distance.

83-85: Please rephrase. Bottlenecks are not a driver of colonization, but can be a consequence of the latter.

Fig. 1c, Fig. S2: I understand that in these PCAs Glasgow may be well separated from the other populations for phylogeographic reasons. The same might apply for Iberian populations. However, there are two observations that seem puzzling and that I would like to better understand. The first is that among Iberian populations only the Portuguese splits off so strongly and neither Madrid nor Barcelona. The second is the spread of Scottish and Portuguese individuals along PC axes 1 and 2. This implies to my understanding that there is actually more diversity found within each of these populations than between the other populations. What could be the explanations for this pattern?

I find it very difficult to get a grasp of the population structure. The authors state that 'the population structure analyses suggest that urban-rural population pairs from the different localities have colonized each urban habitat independently ... , though still a certain number of the urban populations cluster together'. For the nine urban-rural comparisons to be of most use for the purpose advertised in the introduction, indeed the most important question is whether or not urban habitats have been colonized independently. To me, this question is not entirely clearly solved here. One thing that is also striking is the contrast between substantial F_{st} values at a local scale on one hand, and on the other hand admixture analyses that basically do not find any clusters. Moreover, the rural Portuguese individuals seem to be rather consistently intermediate between urban Portuguese individuals and ones from outside Portugal. In summary, I have a hard time getting my head around the results regarding population structure. As these provide an important basis for the main analyses of the paper, I think it is rather vital to resolve this.

106: I do not think it is fair to conclude on a 'overall lack of population structure'. Please see comments above. Please rephrase. What is more, a background of low differentiation rather supports scans for selective sweeps. Thus 'despite' would seem a bit off here.

The correlation of the number of urbanization-associated SNPs with chromosome length is striking and

indeed what we would probably expect in a scenario of polygenic adaptation, because long chromosomes harbour more genes. Still, because short chromosomes have higher gene densities than long ones, I would suggest to add the correlation of urbanization-associated SNPs with the number of genes on a chromosome, as this is what we indeed are interested in.

Fig. S6: Please indicate which population comparisons the panels refer to, and add chromosome names.

177-178: I am not sure to understand this sentence. Is this to mean that 25-50% of outlier windows were found in URBAN populations or that this percentage of windows were outliers? The second would not seem to make much sense. Please rephrase.

I wonder whether the chosen haplotype statistic (R_{sb}) is the most appropriate one to use in the context of this study, especially on its own. Cross-population haplotype statistics predominantly detect completed or close-to-completed sweeps. Is this what we mostly expect between these closely related populations? I would suggest adding iHS or (probably better) nSL to these analyses, as these detect incomplete sweeps as we might rather expect them in geographically close populations that may still exchange a lots of genetic diversity.

480: 'To identify genomic regions showing signs of (incomplete) selective sweeps...' – I think it is misleading to start this section off like this. This is because the authors exclusively use EHS to estimate R_{sb} and only report the latter. Yes, EHS detects incomplete sweeps, but in R_{sb} this effect seems to be cancelled out and R_{sb} serves for a (citation from the paper introducing R_{sb}) 'genomic survey of recent local selective sweeps, especially aimed at those nearly or recently completed' (Tang et al. 2007, Plos Biol).

222: Typo 'f' should read 'of'

I am not sure I agree with the way the authors correlate genetic differentiation with linkage disequilibrium. Rather than estimating the correlation for each population, the authors use PC1 of a PCA on PBS across populations. In my opinion, PC1 will represent the most common signal found in PBS. With rather population-specific signals, this will not well represent differentiation. The same is true for LD (thus in Fig. S11 also LD should be the one of the same population and not LD-PC1). I would strongly suggest performing these analyses separately for each population and removing the PC1 ones, as in my mind these do not make much sense. I think this should result in clearer correlations. If these remain in the negative direction (low differentiation at high LD/low recombination), then this clearly speaks against linked selection as a driver of high differentiation.

Figure S3: To avoid confusion with phylogenetic ML trees, I would suggest referring to these as POPULATION trees.

Reviewer #2 (Remarks to the Author):

I have completed a thorough review of NCOMMS-20-17986 "Repeated genomic signatures of adaptation to urbanization in a songbird across Europe" by Salmón et al (corresponding author Isaksson). In this paper the authors examine whether urbanization has led to repeated parallel selection, background selection or bottlenecks on great tit (Paris major) populations across nine cities selected from across Europe. In total, 192 birds (10-16 birds per urban/rural habitat per city) were genotyped at 526,528 SNPs (post-filtering), and then subject to a wide array of analyses to detect selection across all cities. The authors also use their data to tests for background selection and to infer the absence of demographic bottlenecks using population genetic statistics. After detecting numerous SNPs showing signatures of selection within individual cities, and a handful of cities showing evidence of parallel selection between multiple cities, they infer function of these genes using annotated genomes and corresponding known functions in humans.

I can't emphasize enough how impressed I am with this paper. It is novel, important, rigorous and very well written. The effects of urbanization on evolution is a topic that has only recently emerged as an important topic in evolutionary biology, with a wide array of applications to conservation, sustainability and education. Although the application of genomic techniques to understanding the effects of urbanization on multiple evolutionary processes (e.g. gene flow, drift, selection), has been growing in the last 2-3 years, this is the best paper I am aware of examining how urbanization affects selection genome-wide. Until very recently, there were few studies examining how selection affected natural selection and adaptive evolution, and most studies have focused on heritable phenotypic traits (e.g. Winchell's work) or traits under simple Mendelian inheritance (e.g. Johnson's lab). The handful of previous papers that use genomic techniques to detect selection used either candidate genes (e.g., Mueller et al 2013 Mol Ecology), smaller datasets (e.g. Theodorou et al 2018 Proc B), focused on fewer localities and transcriptomes (e.g. Harris et al 2013 PLoS ONE, Reid et al 2016 Science) or were published just weeks ago (Mueller et al 2020 Mol Ecol). In short the paper is excellent and novel.

I have a few minor suggestions for the authors to consider. The first two are the most important, and the others are of lesser importance.

Line 132 and Fig. S5: While some snps do look qualitatively consistent in direction, many show large variation in magnitude and direction, with evidence of crossing of reaction norms. I suggest performing and adding statistics in the upper right corner of each panel, indicating the significance of the effects of habitat, city and habitat x city effects on allele frequency change.

It would also be nice to see an explicit test supporting the statement: "the directionality and/or magnitude in allele frequency of the 34 identified SNPs showed a highly parallel pattern across all urban populations (Fig. 2f), suggesting that in this species, local adaptation to urban habitats has occurred through repeated shifts in allele frequency of the same loci." If it is provided somewhere and I missed it, please also mention that result here. If it is not in there, a linear mixed effects model should be able to handle this with all alleles treated as random effects, and perhaps also city and city:habitat (with the

idea that these cities are a random subset of all possible cities in Europe) treated as random effects, and the effect of urban/rural habitat treated as a fixed effect in the model; allele frequency would be the response of course.

Line 112: 0.52% of the full SNP dataset showed association with urbanization at an FDR of <1%. Is this not within what we'd expect for a false-positive rate?

Line 123: I suggest revising to read: "...differences in the direction and/or magnitude of allele-frequency change..."

Line 148: Should we be concerned about the low correlation between LFMM and BayPass. Perhaps mention whether this is unusual or typical.

Figure S10: The colour contrast in the legend is rather poor, going from light blue to slightly darker light blue, making it difficult to assess the claim of weak correlations. Consider using a stronger contrast; e.g. white to black, or involving actual change in colours

Line 226: The term "genetic diversity" is used very loosely in multiple places in the paper. In some cases it refers to differentiation between populations (L79), in other cases it is used to refer to heterozygosity (L85) and here it is used to refer to linkage disequilibrium among linked sites. I suggested restricting the term "genetic diversity" for reference to the magnitude of genetic variation within populations, "genetic differentiation" for referring to genetic variation between populations, and never use this term to refer to linkage disequilibrium. LD should be referred to more precisely and some explanation should be made of how it was calculated.

Reviewer #3 (Remarks to the Author):

The authors present an analysis of genetic variation in urban versus rural populations of the great tit (*Parus major*) across nine cities in order to investigate convergence of urban adaptations at the genetic level. Understanding the genomic bases of urban adaptations is currently a very popular topic in the field of urban evolutionary ecology. This study attempts to demonstrate that convergence occurs within the same gene regions, but not the same loci, in some but not all of their populations. Regions of high genetic differentiation between urban-rural pairs appear to be correlated with urbanization, suggesting that urbanization is driving these patterns. Lastly, they identify functional relevance of genomic regions putatively involved in urban adaptation and determine that they mainly fall under the umbrella of neural function and development.

This study addresses a major outstanding question in this field: to what extent are urban adaptations convergent across populations? The breadth of the study is impressive; I am not aware of other studies of this nature that have attempted to understand urban adaptation at such a large scale (nine cities) and the sample sizes per population are modest but appropriate. Overall, I am enthusiastic about this study

but have some significant concerns regarding the conclusions. In addition, I found the text to be difficult at times and the study parts not clearly unified (i.e., how does each portion of the study contribute to the overall story the authors are telling?). I provide more detailed comments below, with minor line-by-line comments following:

(1) Can the authors please explain the decision to use SNP chip over other genomic approaches (e.g., WGS, RAD)? Can the authors please address if the genomic regions targeted were random or specified? Can the authors please address if choice of genomic method may influence results (e.g., ascertainment bias) and if so, how?

(2) There seems to be substantial gene flow between regions and I am not convinced that the urban populations represent independent colonizations from the data presented. This is a substantial issue that I believe needs to be addressed more thoroughly. If there is high gene flow between populations then parallel signatures of selection may not actually reflect convergent adaptive responses. Some other odd patterns to note regarding population structure that need some explanation:

- The Lisbon populations are more divergent from each other than any other population pair. Is there something peculiar about this population pair?
- The Lisbon populations are distinct from all other populations. Any idea why this is the case? Is it possible that this population is hybridizing with another species?
- The admixture plots seem unusual and I'm not really sure what to make of it. This pattern suggests to me that everything outside of Lisbon is panmictic, perhaps with Glasgow representing a third population. If this is true, then the authors have 2-3 populations to test convergence, not 9.
- The PCA of population structure only explains 1.31% in PC1 and 1.17% in PC2 !? These values seem absurdly low. This says to me that the PCA of population structure doesn't tell us much.

I recommend the authors provide more background into the colonization history and relationships between the populations sampled across Europe (e.g., perhaps a haplotype network would be a good start). It might also be informative to analyze shared signatures of selection with respect to population relatedness. In other words, do populations that show similar genomic responses (e.g., Fig. 3) tend to be more closely related to each other? If so this could suggest that either standing genetic variation specific to that lineage is important, or that these urban populations are not independent replicates. There are also methods that enable the researcher to take into account genetic relatedness when testing genotype by habitat relationships that the authors may want to explore. In addition, if the population structure analysis is valid, then I do not see a comparison between all nine cities as valid. Instead, the authors should compare the Lisbon and Glasgow populations to all other populations since the existence of nine populations is not supported.

(3) I'm struggling with the conclusion that the observed patterns of genomic variation indicate local adaptation to urban habitats (e.g., lines 120-121). Natural selection may be producing these patterns,

but other non-adaptive processes may also. This problem is highlighted by the finding that the signatures of selection are shared among subsets of populations and none across all, and that the PBS and Rsb analyses find that most of the selective sweeps are locality-specific. This strongly suggests to me that idiosyncratic non-adaptive population level variation and not parallel natural selection is the mechanism for producing the observed patterns of genetic variation. I would like to see a discussion of these other processes (e.g., genetic drift, gene flow, translocation) and whether it is possible that the genomic variation observed could instead be attributed to non-adaptive processes. There have been some recent papers relevant to this issue (e.g., Miles et al. 2019 Molecular Ecology, Johnson et al. 2018 PRSB).

Line by line comments

Figure 1: What are “main dense urban areas” and how are they defined? This map makes it seem as if the majority of Europe is lacking urbanization. Perhaps an inset zoomed in on each of the cities would be more informative for the reader to understand how urban each of the sites is? Also, since cities are heterogeneous environments, more information on the locations sampled would be useful to the reader to understand the habitats sampled (again, an inset zoom would help here).

Line 47: this isn't entirely accurate. There have been a handful of studies examining convergence in different cities for a variety of taxa, including plants, lizards, fish, and invertebrates. If the authors specifically mean genetic convergence and not phenotypic convergence, then this should be stated. Even so, there are still examples of adaptive genetic convergence in multiple urban populations (thinking specifically of three examples: fish, lizards, clover) that should be acknowledged.

Line 89-92: I don't think your data support this conclusion. The urban and rural separate for Lisbon, and maybe Glasgow, but the other 7 populations are clustered together with no clear separation. Your population structure data suggest a panmictic population not independent colonizations. A haplotype map would help resolve population level relationships. If these truly are not independent colonizations then the conclusion of parallel selection is invalid.

Line 114: I'm not sure I follow why larger chromosomes having more urban-associated SNPs highlights the polygenic nature of urban adaptation. Please explain.

Line 117: variance in PC1 is only 1.98% ? That seems very low to be meaningful.

Line 120-121: Why are these shared SNPs likely involved in local urban adaptation? Other non-adaptive processes could result in these patterns.

Line 124: Please explain this analysis / model. What are the terms? What is the purpose? Which urban-associated SNPs comprise PC1 (the core SNPs?)? Isn't this kind of circular — identifying SNPs associated

with urbanization then asking if they differ by habitat type (urban v non-urban)?

Figure 2: It would be informative if the “core urban SNPs” were indicated in the Manhattan plots (a & b).

Line 174-176: Why was Lisbon used “as the outgroup for estimating PBS”? How sensitive is this analysis to this choice of outgroup? Unclear what the correlation test presented here means, please explain.

Line 185-186: How was “elevated PBS values” determined? Is there a statistical test associated with this? Please clarify.

Line 394: Why was the scale of 1km² chosen? Is this a biologically meaningful scale?

Line 444: How sensitive is this analysis to the choice of the number of latent factors (i.e., other than the two tested)? I don’t follow why the authors chose to test (only) 2 and 4 based on the referenced supplementary materials (it seems that K could also reasonably be 1 or 3).

Reviewer #4 (Remarks to the Author):

The manuscript compares urban and rural populations of great tits to determine whether there are repeated signatures of selection to an urban environment. The manuscript looks for concurrent shifts in allele frequencies among populations as well as selective sweeps. In addition to the interesting question of adaptation to urban environments, which is presumably rapid considering the timeframe during which species adapt, there is large interest in understanding convergent evolution. The findings are potentially novel but not currently presented in a way that is easy to interpret. Moreover, the generalizability of the findings is missing (see comment below about the goals of the paper).

Abstract

The abstract is unclear in whether the replicability is within or between species. Moreover, the two statements about repeated polygenic responses and selective sweeps within the same genes does not jive with the final sentence that adaptation to urbanization has occurred in the same pathways. If selection was occurring in the same genes, it is a much stronger conclusion than same pathways. Therefore, the abstract is a bit misleading about the findings.

Introduction

In the introduction, it would be helpful for the manuscript to include information about how species have phenotypically adapted, as there are examples of the challenges but no evidence of the actual phenotypic adaptations that have occurred, especially because “these adaptations” are referred to (lines 38-41).

There is a substantial debate about convergence vs parallelism and it would be helpful for the

manuscript to define their use of the terms, as parallel and convergent are used seemingly interchangeably.

A question that occurred to me in the introduction is what is the site fidelity for the species? While this is a genomic study, it is important to include this essential information about the study species in the main text, and not solely in the methods. (for example, information around line 378). Also, what is the generation time of great tits?

The goal of the study is not entirely clear, in the abstract and most of the introduction, the manuscript is painted as one studying convergent evolution but at the end of the introduction the manuscript states that the study provides “an outline of genes to target in future ... studies”. I am left with wondering whether the goal is to identify genes to target or a more evolutionary inference endeavor. Of course, these can both be goals of the manuscript but as it is written, the goal(s) are not clear and not written as complementary.

Line 82, the statement “slightly lower” has no statistical support ($P=0.377$) and should be removed.

PCA does not show that pairs are independently colonizing urban and rural habitats. The TreeMix tree would be much more informative with the rural pairs, which would actually show independent colonization and closest pair. There is sufficient data for each population to include in the TreeMix tree.

What was the “control” for LFMM? The 1% FDR cutoff is set by the user and the wide distribution across the genome plus the larger number of chromosomes containing a larger number of SNPs suggests to me just random hits. When imposing a user-selected cutoff, there is always a tail. Perhaps a randomization of the population assignments (PCurb) and re-running to show that the findings are unexpected given the incorrect assignment.

If 2 SNP sets are drawn from the full SNP set, what is the chance they will overlap by 34 SNPs?

Why are the urbanization-associated SNPs indicated with GWAS? GEA may be more appropriate but all of the subscripts on the PCs is a bit unwieldy.

Figure 2f is confusing because minor allele frequency is on the y-axis, yet which population is the MAF measured in? Both populations have $MAF > 0.5$, which is impossible.

How was the overlap in genomic windows determined? (line 164-165)

The title for Figure 3 is a bit misleading because the figure does not show “shared signals of selection”, it shows signals of selection, some of which may be shared.

What is the relevance of the highly negative PBS values? Are those regions that are shared among the rural landscapes?

Were the outliers on the Z-chromosome chosen with the same 99% cutoff as the rest of the genome?

Line 307, the 79 genes were not detailed in the LFMM analysis, however 79 genes were mentioned in the PBS / Fst overlap among 3+ populations.

What is the evidence for recurrent but independent selective sweeps? Same gene but independently in each urban population?

How was the number of sex chromosomes accounted for in the analyses? I did not see a separate analysis of the sex chromosomes in the methods but perhaps I missed it.

Why was a 1km by 1 km rectangle chosen and how does the 1 km x 1km rectangular area related to bird behavior? (line 394).

Fst calculations are described twice in the methods with two different approaches, which was actually used? If two different estimators are used in two different programs, why? And which estimator is used by SNPRelate?

What LD value was used for pruning based on LD with plink? (line 427) some methods details are missing.

The number of SNPs retained and then reported in the filtered dataset differ (line 428).

Do the polygenic findings overlap with the selective sweeps?

In the following document, we present point-by-point response to each reviewer's critique (*italics* typeface), with our responses in **bold** typeface. Note that the line numbers in the responses refer to those in the new version of the manuscript without track changes.

Reviewer #1 (Remarks to the Author):

In this paper, Salmon et al. mined the genomes of nine urban-rural population pairs of Great Tits for genomic regions underlying adaptation to urban environments. The paper present comprehensive analyses that reveal that many independent loci appear to underlie urban adaptation. Nevertheless, the authors also identify genomic regions in which selective sweeps occurred in several urban populations. The identified genomic regions are predominantly related to neutral function and development.

I think this is an interesting paper that could make for a nice contribution to Nature Communications. Nevertheless, I have a couple of comments that are in part related to presentation and in part to analysis and interpretation. My main comment is probably with regard to the analysis of population structure. These do in my opinion not provide a clear picture of how populations are related; notably there are a result or two that need explaining. Please see detailed comments.

We thank the reviewer for the thoughtful and constructive comments and have addressed all critiques in our extensive revision, particularly regarding the analysis of the population structure. Please find our detailed replies below.

Detailed comments

25-26: Written as is, it is unclear not entirely clear what 'localized' means and why this hampers replicability. Please rephrase.

Thank you for this comment, we referred to spatial replication. We have now thoroughly modified the abstract in order to gain clarity.

29: Are these sweeps independent or the same, and what is the evidence?

We have revised the abstract and cautioned our previous statement and rephrase it to: "...We found that a combination of parallel polygenic allele frequency shifts and recurrent but largely independent selective sweeps are associated with the adaptation of great tits to urban environments. While haplotypes under selection were rarely shared across urban populations, selective sweeps occurred within the same genes, mostly linked to neural function and development..." As described later in the manuscript (e.g., L402-413) we infer that sweeps associated with the same genes likely occurred independently as sweep haplotypes and the position of sweeps are not, or only rarely, shared across urban populations.

29-32: It is not entirely clear how the association with the two specific functions points to adaptation through the same pathways. Please rephrase.

We agree that this sentence was not very clear, and we have rephrased it within the abstract word limit: "...While haplotypes under selection were rarely shared across urban populations, selective sweeps occurred within the same genes, mostly linked to

neural function and development. Collectively, we show that repeated urban adaptation in a widespread songbird occurred through unique and parallel selective sweeps in a core-set of behaviour-linked genes. “

44: *Evolutionary signals of what? What are ‘evolutionary signals’? What are ‘signals of adaptation’ (45)?*

We agree that this was a vague statement and we have thoroughly worked on the introduction to make it overall clearer and more accurate, avoiding statements like that (L37-43).

49/50: *Written as is, this kind of takes for granted that there is convergent evolution. I would suggest introducing this differently. In my mind, the first question is whether there is parallel/convergent evolution at all and what its origins are.*

We agree with the reviewer that the first question should be if there is parallel evolution. Thus, we have revised this section in the introduction of the manuscript, so we better introduce parallelism in the genomic response to urban habitats and its importance for understanding the adaptation to urbanisation (L48-64).

52: *Please delete ‘strong’*

Thank you for this we agree with the reviewer’s suggestion and we have avoided the use of the word “strong” and modify the sentence accordingly (L51-53).

54: *Here the authors write about ‘parallel evolutionary adaptation’ (versus convergent evolution before). It would in my opinion be important to stick with one and clearly define what the definition entails, as there are different definitions of each.*

This is a good point and we apologise for the confusion produced when using interchangeably both terminologies. In this new submission we have stick to “*parallel evolutionary adaptation*” throughout the manuscript. We base this on our understanding of “*parallel evolution*” as the process of independent and replicated evolution of similar adaptations/phenotypes in closely related populations or species (see e.g. Elmer & Meyer 2011 Trends Ecol. Evol., cited in the text). This has been defined across the introduction and text (L48-64; L73-77; L352-357).

I have the feeling that the way parallel/convergent evolution is presented here mingles two things: The prevalence of the latter per se, and parallel responses to selection as a tool to confirm findings in one population with the same findings in other populations. It is unclear to me, which of these the authors address in first line. I would suggest adapting the introduction accordingly.

Thank you for pointing this out. We have adjusted the introduction accordingly (L48-64). We are now clearly pointing out that we are using the presence of (repeated) adaptation to cities to understand if the genomic responses to selection are parallel or not, and the possible mechanisms underneath, see e.g. (study objectives L71-77).

69: *What was the goal of these two approaches?*

We are sorry for not being more explicit here before. The rationale of these two approaches is to capture different aspects of local adaptation:

1) We used genotype-environment association (GEA) to test for more “polygenic” signals of urban adaptation, including subtle allele frequency shifts in many loci across urban populations (e.g. parallel adaptation).

2) The analyses of selective sweeps is used to detect signatures of selection in individual urban populations that might not show up in the GEA, as sweeps might be restricted to single populations (e.g. population-specific responses).

We have now stated both aspects more clearly in the introduction (L74-77) and again when discussing and presenting the results (L237-245).

80: *F_{st}* values of up to 5% at such small geographic distance do not represent low differentiation at such a local scale. If significantly different from zero, even smaller values would seem rather substantial for passerine birds that usually show high dispersal rates. It would therefore seem important to test whether differentiation is different from zero. Moreover, how does local differentiation compare to differentiation at the continent level? When I look at the values, it would seem that differentiation at the local scale is comparable to differentiation between populations. The question is in this case indeed, whether these values are significantly different from zero, if there is not the least pattern of isolation by distance.

We have now tested how differentiation on the local level corresponds to differentiation on the continent-wide scale and observed some interesting patterns (Figs. 2c and d in the revised manuscript, see here for reference).

(i) We detect a signature of isolation-by-distance across populations, which is stronger when only analysing urban populations compared to rural populations. We interpret this signal as evidence for limited dispersal of urban compared to rural birds (L144-148).

(ii) Second, local-scale differentiation is comparable to levels of genetic differentiation between urban populations (urban vs urban), and significantly higher than the average differentiation between rural populations (rural vs rural). This could be either explained by limited gene flow, due to strong local adaptation, or non-adaptive processes. However, under a non-adaptive scenario, we would expect an increase in differentiation at the local scale (urban vs rural from the same population pair) with geographic distance between them. Yet, we did not detect any correlation between local-scale distance and differentiation, which we interpret as evidence for local adaptation as a driver of the differentiation.

Lastly, we used a pairwise permutation approach based on a thinned SNP dataset (for computational reasons, ~ 1 SNP per 50kb) to test if differentiation is significant from zero on all scales. All pairwise comparisons, except one between the urban Munich and rural Paris populations were significant in the analysis ($P < 0.05$). Therefore, even though

population structuring appears weak there seem to be significant differences in structure between geographically close populations, as supported by the *EEMS* and *UMAP* analysis.

83-85: Please rephrase. Bottlenecks are not a driver of colonization, but can be a consequence of the latter.

Thank you for this comment, we have re-written the section accordingly (L114-117).

Fig. 1c, Fig. S2: I understand that in these PCAs Glasgow may be well separated from the other populations for phylogeographic reasons. The same might apply for Iberian populations. However, there are two observations that seem puzzling and that I would like to better understand. The first is that among Iberian populations only the Portuguese splits off so strongly and neither Madrid nor Barcelona. The second is the spread of Scottish and Portuguese individuals along PC axes 1 and 2. This implies to my understanding that there is actually more diversity found within each of these populations than between the other populations. What could be the explanations for this pattern?

*I find it very difficult to get a grasp of the population structure. The authors state that ‘the population structure analyses suggest that urban-rural population pairs from the different localities have colonized each urban habitat independently ... , though still a certain number of the urban populations cluster together’. For the nine urban-rural comparisons to be of most use for the purpose advertised in the introduction, indeed the most important question is whether or not urban habitats have been colonized independently. To me, this question is not entirely clearly solved here. One thing that is also striking is the contrast between substantial *Fst* values at a local scale on one hand, and on the other hand admixture analyses that basically do not find any clusters. Moreover, the rural Portuguese individuals seem to be rather consistently intermediate between urban Portuguese individuals and ones from outside Portugal. In summary, I have a hard time getting my head around the results regarding population structure. As these provide an important basis for the main analyses of the paper, I think it is rather vital to resolve this.*

Thank you for these thoughtful comments. We agree that a better understanding of the population structure is important for the correct interpretation of the results and we have therefore conducted new and more thorough population structure analyses, including new analyses that are more appropriate for spatial analysis than *Admixture*.

(i) Regarding the split and spread of individuals from Glasgow and Lisbon, there are a few potential explanations. First, Glasgow and Lisbon, particularly the urban populations, show a reduced heterozygosity compared to all other populations (although in general heterozygosity is not reduced in urban populations), likely due to the fact that these populations are located at the range edge of the Great tit distribution in Europe. Such reduced heterozygosity can lead to distinct patterns in population structure analyses and give the impression of more diverged populations. Furthermore, variation in individual heterozygosity within populations can lead to a wider spread across the PCA axis. These patterns are more pronounced when each eigenvector in a PCA only explains a small amount of variation, as it is the case here. In our study, each eigenvector indeed only explains a relatively small amount of variation and differences between eigenvectors are also small. In this new revised version of the manuscript we have summarized the variation of the first 20 PCAs into two dimensions using a *UMAP* approach. We chose the

first 20 PCAs based on the distribution of variance explained, although the same analysis using the first 10 PCAs gave similar results (Fig. 2a). This approach, which summarises a large amount of the genetic variation, still indicates a stronger divergence of Glasgow and Lisbon from the other populations, but an overall reduced spread within populations and a strong divergence between the urban and rural populations of Lisbon. Moreover, the revised *TreeMix* analysis (Fig. 2b), further suggests increased drift in the urban populations from Glasgow and Lisbon, with the highest drift parameter for urban Lisbon. In the light of these results, we suggest that the increased divergence in those particular populations (Glasgow and Lisbon) is driven by a reduced heterozygosity in the urban populations. A potential mechanism underneath might be introgression in the urban Lisbon population, but in that scenario, we would have also expected an increase rather than a decrease in diversity. Therefore, we conclude that a smaller population size and increased drift at the urban range edge are the more likely explanations.

(ii) Regarding the independent colonisation of urban habitats, the *UMAP* and revised *TreeMix* analyses shows that in all cases but Munich and Paris, urban populations are divergent from each other and either cluster with their respective rural population (Madrid, Gothenburg, Barcelona, Glasgow) or cluster separately from their rural and other populations (Lisbon, Malmö, Milan). Munich and Paris do not show any distinct separation, which is also confirmed but the indistinct clustering in the *TreeMix* tree (Fig. 2b). This clustering of Paris and Munich is largely explained by relatively strong effective gene flow in the middle of our sampling range (*EEMS* analysis, Fig. 2e). In addition, the *EEMS* analysis further revealed reduced gene flow between populations in the other parts of Europe, even between relatively close populations, such as GOT (Gothenburg) and MAL (Malmö). Overall, gene flow is more strongly reduced between urban compared to rural populations, as indicated by weaker isolation-by-distance between rural birds compared to urban ones (mantel's $r_{\text{urban}} = 0.46$ vs $r_{\text{rural}} = 0.43$), potentially providing an additional analyses for a stronger separation of the urban compared to the rural Lisbon population.

(iii) Regarding the low separation in the previous Admixture analysis, which has been now removed from the current version. We have opted to present instead *fastStructure* analysis (Fig. S2), although we agree that these are generally not well suited for spatial analyses with isolation-by-distance and relatively weak divergence patterns. Nonetheless, we have included the revised *fastStructure* analysis for comparison, but refer in the manuscript to the *UMAP*, *EEMS* and *IBD* analyses as these are better descriptors of the population structure in European great tits. For additional information and exploration of the studied species population structure we also refer to the great tit HapMap pre-print (Spurgin et al 2019 bioRxiv: <https://www.biorxiv.org/content/10.1101/561399v1>).

Overall, based on these results, we assume that we have 8 distinct urban-rural population pairs. However, because of the fact that genetic differentiation is still present between Munich and Paris, and the selection pressures might differ between localities we rather prefer to analyse them as 9 urban-rural population pairs. Indeed, we believe that analysing Munich and Paris together would potentially lead us to miss location-specific signatures of selection and overestimate genetic parallelism.

106: I do not think it is fair to conclude on a 'overall lack of population structure'. Please see

comments above. Please rephrase. What is more, a background of low differentiation rather supports scans for selective sweeps. Thus 'despite' would seem a bit off here.

Thank you for this, we have now removed that introductory sentence in the new version (L163-165). Indeed, we agree that the weak population structuring actually aids us in correctly identify selective sweeps.

The correlation of the number of urbanization-associated SNPs with chromosome length is striking and indeed what we would probably expect in a scenario of polygenic adaptation, because long chromosomes harbour more genes. Still, because short chromosomes have higher gene densities than long ones, I would suggest to add the correlation of urbanization-associated SNPs with the number of genes on a chromosome, as this is what we indeed are interested in.

Indeed, we agree that the number of genes would be a better proxy for the number mutational targets. We have now added this analysis in the new version (Fig. 3b). We detect a strong correlation between the number of associated SNPs and the number of genes per chromosome ($R^2=0.90$), supporting a scenario of polygenic adaptation.

Fig. S6: Please indicate which population comparisons the panels refer to, and add chromosome names.

We have modified Fig. S6 and included population comparisons and chromosome names (see Fig. S10).

177-178: I am not sure to understand this sentence. Is this to mean that 25-50% of outlier windows were found in URBAN populations or that this percentage of windows were outliers? The second would not seem to make much sense. Please rephrase.

We apologise for the unclear description. We have removed some analyses and therefore this section is not part of the revised manuscript.

I wonder whether the chosen haplotype statistic (R_{sb}) is the most appropriate one to use in the context of this study, especially on its own. Cross-population haplotype statistics predominantly detect completed or close-to-completed sweeps. Is this what we mostly expect between these closely related populations? I would suggest adding iHS or (probably better) nSL to these analyses, as these detect incomplete sweeps as we might rather expect them in geographically close populations that may still exchange a lots of genetic diversity.

Thank you for the suggestion. We agree that the focus on R_{sb} will likely miss some more recent or ongoing selective sweeps. We have now included a newly developed cross-population nSL analysis ($XP-nSL$) as implemented in *selscan* v.2.2 (Szpiech et al 2020, bioRxiv: <https://www.biorxiv.org/content/10.1101/2020.05.19.104380v1>). In this new analysis we find a large number of new genes associated with ongoing selection/with incomplete sweeps (Fig. 4a). Therefore, we have revised related analyses and interpretation accordingly, but the overall conclusions remain the same: *i*) sweep haplotypes are often population-specific (not shared) but *ii*) affect the same gene in multiple populations and *iii*) many of those genes are involved in behaviour and neural development/control (e.g., L417-446).

480: 'To identify genomic regions showing signs of (incomplete) selective sweeps...' – I think it is misleading to start this section off like this. This is because the authors exclusively use

EHHS to estimate R_{sb} and only report the latter. Yes, EHHS detects incomplete sweeps, but in R_{sb} this effect seems to be cancelled out and R_{sb} serves for a (citation from the paper introducing R_{sb}) ‘genomic survey of recent local selective sweeps, especially aimed at those nearly or recently completed’ (Tang et al. 2007, Plos Biol).

We have revised this section and have added $XP-nSL$ as an additional way for detecting incomplete and complete selective sweeps (L246-247; see Methods: “Haplotype-based selection analyses”: L628-646). See also previous response for further details.

222: Typo ‘f’ should read ‘of’

Thank you for finding this typo, although in the new version the sentence has been re-worded, and it is not applicable anymore.

I am not sure I agree with the way the authors correlate genetic differentiation with linkage disequilibrium. Rather than estimating the correlation for each population, the authors use PC1 of a PCA on PBS across populations. In my opinion, PC1 will represent the most common signal found in PBS. With rather population-specific signals, this will not well represent differentiation. The same is true for LD (thus in Fig. S11 also LD should be the one of the same population and not LD-PC1). I would strongly suggest performing these analyses separately for each population and removing the PC1 ones, as in my mind these do not make much sense. I think this should result in clearer correlations. If these remain in the negative direction (low differentiation at high LD/low recombination), then this clearly speaks against linked selection as a driver of high differentiation.

We agree that our approach mostly focuses on broader patterns in PBS or other statistics. Therefore, in the revised manuscript we have followed the reviewer’s advice and performed population-specific comparisons for LD [measured in urban populations] with our main selection/differentiation statistics in non-overlapping 200kb windows (F_{ST} , $XP-nSL$, R_{sb}). Additionally, we have included intron-based GC-content (summarised in non-overlapping 200kb windows) as an additional proxy for long-term recombination rate variation across the genome (e.g., Charlesworth et al. 2020 Mol. Biol. Evol.). High recombination regions harbour a higher GC content. LD and GC-content are consistently (negatively) correlated, which indicates that they represent similar signals.

Nonetheless, we did not detect a consistent pattern across populations (Fig. 4c) and most of the effect sizes are low (Pearson correlation coefficients) and their distributions overlap with zero (Fig. 4c), indicating inconsistent correlations between summary statistics. Under a scenario of background selection as the main driver of divergence, we would expect a consistent positive correlation between LD/GC and the measures of differentiation/selection.

We interpret this lack of consistency and weak correlations as an evidence for a more important role of positive selection, opposite to linked selection/low recombination rates, as the driver of selection and differentiation patterns in our study.

Figure S3: To avoid confusion with phylogenetic ML trees, I would suggest referring to these as POPULATION trees.

We agree and have revised it accordingly.

Reviewer #2 (Remarks to the Author):

I have completed a thorough review of NCOMMS-20-17986 "Repeated genomic signatures of adaptation to urbanization in a songbird across Europe" by Salmón et al (corresponding author Isaksson). In this paper the authors examine whether urbanization has led to repeated parallel selection, background selection or bottlenecks on great tit (Paris major) populations across nine cities selected from across Europe. In total, 192 birds (10-16 birds per urban/rural habitat per city) were genotyped at 526,528 SNPs (post-filtering), and then subject to a wide array of analyses to detect selection across all cities. The authors also use their data to tests for background selection and to infer the absence of demographic bottlenecks using population genetic statistics. After detecting numerous SNPs showing signatures of selection within individual cities, and a handful of cities showing evidence of parallel selection between multiple cities, they infer function of these genes using annotated genomes and corresponding known functions in humans.

I can't emphasize enough how impressed I am with this paper. It is novel, important, rigorous and very well written. The effects of urbanization on evolution is a topic that has only recently emerged as an important topic in evolutionary biology, with a wide array of applications to conservation, sustainability and education. Although the application of genomic techniques to understanding the effects of urbanization on multiple evolutionary processes (e.g. gene flow, drift, selection), has been growing in the last 2-3 years, this is the best paper I am aware of examining how urbanization affects selection genome-wide. Until very recently, there were few studies examining how selection affected natural selection and adaptive evolution, and most studies have focused on heritable phenotypic traits (e.g. Winchell's work) or traits under simple Mendelian inheritance (e.g. Johnson's lab). The handful of previous papers that use genomic techniques to detect selection used either candidates genes (e.g., Mueller et al 2013 Mol Ecology), smaller datasets (e.g. Theodorou et al 2018 Proc B), focused on fewer localities and transcriptomes (e.g. Harris et al 2013 PLoS ONE, Reid et al 2016 Science) or were published just weeks ago (Mueller et al 2020 Mol Ecol). In short the paper is excellent and novel.

Thank you very much for your very constructive review and we are glad you found our paper novel. Please find our detailed responses below.

I have a few minor suggestions for the authors to consider. The first two are the most important, and the others are of lesser importance.

Line 132 and Fig. S5: While some snps do look qualitatively consistent in direction, many show large variation in magnitude and direction, with evidence of crossing of reaction norms. I suggest performing and adding statistics in the upper right corner of each panel, indicating the significance of the effects of habitat, city and habitat x city effects on allele frequency change.

Thank you very much for your suggestion, we have now estimated the effect sizes for each individual associated SNP including the "core urbanisation SNPs" and have plotted their effect sizes in Fig. 3d. These results highlight that for three quarters of the "core urbanisation SNPs" the parallel effect of "Habitat" is stronger than the non-parallel "Habitat x Locality") effect, supporting the parallelism in allele frequency shift and trajectory (Fig S7). We have also included the effect sizes in the revised supplementary

Fig. S9. Furthermore, we have also estimated the “directionality” of allele frequency shifts, meaning we have determined for all urbanisation-associated SNPs (*LFMM* and “*core urbanisation SNPs*”) if the same allele increased in frequency in the urban compared to the adjacent rural population. This further highlight that within the “*core urbanisation SNPs*”, same allele increased in frequency occurs in at least 7 populations (Fig. S6c).

It would also be nice to see an explicit test supporting the statement: "the directionality and/or magnitude in allele frequency of the 34 identified SNPs showed a highly parallel pattern across all urban populations (Fig. 2f), suggesting that in this species, local adaptation to urban habitats has occurred through repeated shifts in allele frequency of the same loci." If it is provided somewhere and I missed it, please also mention that result here. If it is not in there, a linear mixed effects model should be able to handle this with all alleles treated as random effects, and perhaps also city and city:habitat (with the idea that these cities are a random subset of all possible cities in Europe) treated as random effects, and the effect of urban/rural habitat treated as a fixed effect in the model; allele frequency would be the response of course.

We are sorry to hear that this was not clear in the previous version. Similar to all urbanisation-associated SNPs in the *LFMM* model, we had also used linear models with PC1 for shared loci as the response variable to determine the effect size of “*Habitat*” (parallel urban vs rural allele frequency divergence), “*Locality*” (evolutionary history) and “*Locality x Habitat*” (non-parallel allele frequency divergence) on patterns of allele frequency divergence (summarised as PC1 across all 34 “*core urbanisation SNPs*”, Figs. 3c and S6b and d) (following e.g., Stuart et al. 2017 Nat. Ecol. Evol.; Bolnick et al. 2018 Annu. Rev. Ecol. Evol. Syst.). This model suggested that habitat has by far the strongest effect on allele frequency changes, explaining 73% of the total variation compared to “*Locality*” (19%) and “*Habitat x Locality*” (13%) (L218-222; Fig. S6d). Furthermore, as outlined above, we have also included the same models for all individual urbanisation-associated SNPs (*LFMM* and “*core urbanisation SNPs*”) with a focus on “*core urbanisation SNPs*” (Figs. 3d and S6b). These analyses further evidence, that particularly for the “*core urbanisation SNPs*”, the parallel “*Habitat*” effect is stronger than the non-parallel “*Locality x Habitat*” interaction effect, supporting the statement that allele frequency shifts are mostly parallel across populations in the detected SNPs.

We have also analysed in how many urban populations the same allele increased in frequency. We found that compared to all *LFMM*-SNPs (2,757 SNPs) the 34 “*core urbanisation SNPs*” show a higher proportion of parallel shifts in all populations (Figs 3d and S6b) and in general shifted in the same direction in the majority of urban populations, supporting the conclusion of local adaptation through repeated shifts in allele frequency of the same loci.

Line 112: 0.52% of the full SNP dataset showed association with urbanization at an FDR of <1%. Is this not within what we'd expect for a false-positive rate?

If we are not completely wrong, in our context the false-discovery rate (FDR), i.e., the proportion of Type I errors among rejected null hypotheses (Korthauer et al. 2019 Genome Biol.), gives the number of urbanisation-associated SNPs [here 2,758 SNPs] that have been wrongly associated with urbanisation (i.e., false positives). Therefore, using a FDR of 1% (q-value < 0.01) we would expect that 1% of all our significantly associated loci with a q-value < 0.01, which is ~28 SNPs (or 0.0053% of the entire dataset), are false-

positives. To further support our interpretation, we have also tested a permutation approach for determining outlier loci. However, we found this too liberal (see Fig. S4a) and therefore kept the results from the FDR correction.

Line 123: I suggest revising to read: "...differences in the direction and/or magnitude of allele-frequency change..."

Thank you for the suggestion, we have now revised that whole section (L199-226).

Line 148: Should we be concerned about the low correlation between LFMM and BayPass. Perhaps mention whether this is unusual or typical.

We are afraid that low correlations between Genotype-Environment-Association (GEA) methods are relatively common, i.e., between LFMM and BayPass, largely due to different assumptions, different effects of demographic history, and different levels of correction for e.g. population structure. Several papers have explored this question in more detail (e.g., Forester et al. 2018 Mol. Ecol.) and overall GEA analyses are a rapidly developing field that is currently still lacking behind other GWAS approaches.

In particular, BayPass corrects more strongly for population structure, leading to lower numbers of genotype-environment associations, as local adaptation is often correlated to local population structure, leading to many false-negatives (Forester et al. 2018 Mol. Ecol.). On the other hand, more lenient approaches, such as LFMM, might lead to more false-positives, yet as our post-hoc analysis shows, the overall signal of urban-habitat split is very strong (e.g., Figs. 3c and d). Additionally, the 34 "core urbanisation SNPs" we identified, are most likely true-associations that are highly parallel across urban-rural population pairs, which is supported by our inference of highly parallel allele frequency changes in most of those shared SNPs (Figs. 3d, S7 and S8).

Figure S10: The colour contrast in the legend is rather poor, going from light blue to slightly darker light blue, making it difficult to assess the claim of weak correlations. Consider using a stronger contrast; e.g. white to black, or involving actual change in colours

We agree that the contrast was not optimal. We have replaced this figure and all correlations are now shown in Figs. 5b and c.

Line 226: The term "genetic diversity" is used very loosely in multiple places in the paper. In some cases it refers to differentiation between populations (L79), in other cases it is used to refer to heterozygosity (L85) and here it is used to refer to linkage disequilibrium among linked sites. I suggested restricting the term "genetic diversity" for reference to the magnitude of genetic variation within populations, "genetic differentiation" for referring to genetic variation between populations, and never use this term to refer to linkage disequilibrium. LD should be referred to more precisely and some explanation should be made of how it was calculated.

We agree with the reviewer and we have corrected the language in the revised manuscript following the reviewer's recommendations, in particular through the section in the Results and discussion: "Evolutionary drivers of genetic differentiation and signatures of selection". LD calculations are now clearly stated in the Methods section (672-675)

Reviewer #3 (Remarks to the Author):

*The authors present an analysis of genetic variation in urban versus rural populations of the great tit (*Parus major*) across nine cities in order to investigate convergence of urban adaptations at the genetic level. Understanding the genomic bases of urban adaptations is currently a very popular topic in the field of urban evolutionary ecology. This study attempts to demonstrate that convergence occurs within the same gene regions, but not the same loci, in some but not all of their populations. Regions of high genetic differentiation between urban-rural pairs appear to be correlated with urbanization, suggesting that urbanization is driving these patterns. Lastly, they identify functional relevance of genomic regions putatively involved in urban adaptation and determine that they mainly fall under the umbrella of neural function and development.*

This study addresses a major outstanding question in this field: to what extent are urban adaptations convergent across populations? The breadth of the study is impressive; I am not aware of other studies of this nature that have attempted to understand urban adaptation at such a large scale (nine cities) and the sample sizes per population are modest but appropriate. Overall, I am enthusiastic about this study but have some significant concerns regarding the conclusions. In addition, I found the text to be difficult at times and the study parts not clearly unified (i.e., how does each portion of the study contribute to the overall story the authors are telling?). I provide more detailed comments below, with minor line-by-line comments following:

(1) Can the authors please explain the decision to use SNP chip over other genomic approaches (e.g., WGS, RAD)? Can the authors please address if the genomic regions targeted were random or specified? Can the authors please address if choice of genomic method may influence results (e.g., ascertainment bias) and if so, how?

We agree that we should have provided a better rationale for this in the text and now we have included some further information (L513-526). The main decision on using this particular SNP chip has been described in Kim et al 2018 Mol. Ecol. Briefly, the used SNP chip has showed that the majority of target SNPs could be genotyped reliably and accurately across multiple Great tit populations and generate several hundreds of SNP genotypes with considerably less than 1% typing error. Moreover, the SNP chip has a relatively low cost per SNP per individual and it is relatively robust to low yield or highly degraded DNA, whereas the DNA requirements for whole-genome sequencing technologies tend to be more demanding. Finally, other options could be more technically demanding, in terms of post-sequencing processing of data and also, this SNP chip has already been successfully used to detect signatures of selection (Bosse et al. 2017 Science), GWA studies (Bosse et al. 2017 Science; Gienapp et al 2017 Front. Genet.) or the role of CNVs on genomic architecture (da Silva et al 2018 BMC Genomics).

It is true that the ascertainment bias could be a problem, although in principle it should be small as the SNP discovery for the chip was performed using individuals from multiple populations across Europe (Kim et al. 2018 Mol. Ecol.). Furthermore, the relatively weak population structuring and strong gene flow across Europe further reduces the risk of a strong ascertainment bias.

In general, SNP chips or other genotyping arrays can lead to an overestimation of heterozygosity, although the relative level of H_e between populations should not be affected, and F_{ST} can be weakly underestimated, particularly for low F_{ST} values (e.g., explored in Malomane et al. 2018 BMC Genomics). On the other hand, PCA estimates are usually not strongly impacted by the use of SNP chips. However, if strong ascertainment bias exists, these estimates can diverge even more strongly from the “true” value, but as argued above, we do not expect that to be the case here.

(2) There seems to be substantial gene flow between regions and I am not convinced that the urban populations represent independent colonizations from the data presented. This is a substantial issue that I believe needs to be addressed more thoroughly. If there is high gene flow between populations then parallel signatures of selection may not actually reflect convergent adaptive responses. Some other odd patterns to note regarding population structure that need some explanation:

- The Lisbon populations are more divergent from each other than any other population pair. Is there something peculiar about this population pair?

- The Lisbon populations are distinct from all other populations. Any idea why this is the case? Is it possible that this population is hybridizing with another species?

- The admixture plots seem unusual and I'm not really sure what to make of it. This pattern suggests to me that everything outside of Lisbon is panmictic, perhaps with Glasgow representing a third population. If this is true, then the authors have 2-3 populations to test convergence, not 9.

- The PCA of population structure only explains 1.31% in PC1 and 1.17% in PC2 !? These values seem absurdly low. This says to me that the PCA of population structure doesn't tell us much.

I recommend the authors provide more background into the colonization history and relationships between the populations sampled across Europe (e.g., perhaps a haplotype network would be a good start). It might also be informative to analyze shared signatures of selection with respect to population relatedness. In other words, do populations that show similar genomic responses (e.g., Fig. 3) tend to be more closely related to each other? If so this could suggest that either standing genetic variation specific to that lineage is important, or that these urban populations are not independent replicates. There are also methods that enable the researcher to take into account genetic relatedness when testing genotype by habitat relationships that the authors may want to explore. In addition, if the population structure analysis is valid, then I do not see a comparison between all nine cities as valid. Instead, the authors should compare the Lisbon and Glasgow populations to all other populations since the existence of nine populations is not supported.

Thank you for the really constructive comments. We have revised the population structure analyses and added additional analyses to reconstruct more precisely the population structure in spatial context (EEMS, Isolation-by-distance) and combine information from more principal component axes into 2 dimensions (UMAP) (Margaryan et al. 2020 Nature). These have helped to better resolve the population structure and provided a more detailed and slightly different picture compared to the Admixture analyses (see response to Reviewer #1 for details). In brief, the UMAP analyses (Fig. 2a) has highlighted that most urban populations, except Munich and Paris, form distinct genetic clusters that either tightly cluster with their respective rural pair, or form

completely separate clusters, indicating their independent history. The independent colonisation history of urban habitats is also supported by the revised *TreeMix* population tree that show the clustering of urban populations with adjacent rural populations, except in Munich and Paris for which the relationships are ambiguous due to strong gene flow. Strong gene flow between Munich and Paris, but barriers to gene flow between other cities, was confirmed by estimates of the effective migration surface (*EEMS*; Petkova et al. 2016 Nat. Genetics). However, as selection pressures and/or localised genomic responses to urbanisation in Munich and Paris are still potentially different, we analysed these pairs also separately.

The genetic differentiation of populations was further highlighted by significant isolation-by-distance patterns, which were stronger for urban populations. In general, the urban populations showed stronger differentiation than rural ones, which suggests reduced gene flow between urban habitats.

Overall, our revised analyses indicate the presence of population structure across Europe, which is stronger for urban populations, and therefore we hypothesise that urban habitats were likely colonised independently from a more strongly admixed rural “meta-population” (L153-160).

Regarding the question if more closely related populations show more similar genomic responses to urbanisation, we have found no indication for this. For example, comparing the number of shared outlier windows associated with selective sweeps to the genetic (mantel's $r = 0.28$, $p=0.15$) or geographic distance (mantel's $r = 0.05$, $p=0.40$) between populations, did not show any significant correlations (see figures below). The correlation between number of shared sweep genes with geographic distance was also non-significant, albeit stronger (mantel's $r = 0.26$, $p=0.08$). Thus, we do not think that gene flow/introgression of adaptive genes are a major component on the urban adaptation in the studied populations.

(3) I'm struggling with the conclusion that the observed patterns of genomic variation indicate local adaptation to urban habitats (e.g., lines 120-121). Natural selection may be producing these patterns, but other non-adaptive processes may also. This problem is highlighted by the finding that the signatures of selection are shared among subsets of populations and none across all, and that the PBS and Rsb analyses find that most of the

selective sweeps are locality-specific. This strongly suggests to me that idiosyncratic non-adaptive population level variation and not parallel natural selection is the mechanism for producing the observed patterns of genetic variation. I would like to see a discussion of these other processes (e.g., genetic drift, gene flow, translocation) and whether it is possible that the genomic variation observed could instead be attributed to non-adaptive processes. There have been some recent papers relevant to this issue (e.g., Miles et al. 2019 Molecular Ecology, Johnson et al. 2018 PRSB).

Thank you for the constructive comments. We have included new analyses to assess the possible roles of genetic drift, recombination/background selection and gene flow in driving the observed genomic patterns. We have further revised the text and analyses accordingly to address the reviewer's concerns. We have summarised the additional evidence for the role of natural selection rather than non-adaptive processes below:

(1) Regarding the role of genetic drift and gene flow, we would expect that a reduction in gene flow due to urbanisation and thus increased drift, would be manifested by reduced genetic diversity within urban habitats. Yet, as stated in the manuscript, there is on average no significant difference in H_e between urban and rural populations (Table S1, L114-117). Furthermore, if reduced gene flow (e.g., through reduced dispersal) and increased drift were a main driver of these variable patterns, we would have expected an increase in genetic differentiation with distance between the sampled urban and rural populations (isolation-by-distance). However, there was no correlation ($R^2 = -0.11$, $P=0.645$; Fig. 4e), further supporting that other factors are driving these variable genomic patterns.

(2) If variation in background selection, e.g., in low recombination regions, was a major driver of genomic differentiation between urban and rural great tits due to the reduction of genetic diversity in those genomic regions, we would expect that:

(i) these patterns are conserved across cities as large-scale recombination rates will be similar.

(ii) that patterns of differentiation and selection in genomic windows are not significantly or consistently associated with recombination rate.

As we did not have access to direct recombination rate estimates for great tits, we used the intronic GC-content (Charlesworth et al. 2020 Mol. Biol. Evol.) and LD estimates as proxies for recombination rate. Yet, patterns of differentiation/selection were not correlated with these recombination proxies, suggesting that variation in recombination and background selection are not the main driver of genomic landscapes (L322-335; Fig. 4c).

(3) Lastly, we don't think that natural selection necessarily has to lead to highly parallel genomic signatures of selection across populations. This expectation only holds when:

(i) phenotypes under selection are simple oligogenic traits (e.g., the production of hydrogen cyanide in clover, see for example Johnson et al 2018 Proc. R. Soc. B). Traits with a polygenic genetic basis are more likely experiencing higher genetic redundancy/reduced genetic constrained (e.g., Barghi et al. 2020 Nat. Rev. Gent.). Thus, the same or different

fitness optima can be reached via a wider range of genomic routes, leading to lower genomic parallelism.

(ii) The amount of standing genetic variation in a population also impacts the level of genomic parallelism, with more standing genetic variation often reducing parallelism (e.g., Zheng et al. 2019 Science).

The large number of associations with urbanisation, the number of genetic signatures of differentiation/selection across the genome in individual populations, ranging from 362 to 1,366 genes per population, and the variable amount of outlier sharing suggest that the genomic response to urbanisation is more likely polygenic.

Overall, we therefore conclude that divergent selection between urban and rural great tits is the main driver for the divergent genomic landscapes and that the variable level of parallelism across populations (highly parallel shifts in some urbanisation-associated SNPs to low sharing of selective sweeps [although still more than expected by chance]) is a result of the fact that adaptation to urbanisation in great tits, e.g. behavioural shifts, are polygenic traits and thus are more likely associated with higher genetic redundancy/lower parallelism.

Line by line comments

Figure 1: What are “main dense urban areas” and how are they defined? This map makes it seem as if the majority of Europe is lacking urbanization. Perhaps an inset zoomed in on each of the cities would be more informative for the reader to understand how urban each of the sites is? Also, since cities are heterogeneous environments, more information on the locations sampled would be useful to the reader to understand the habitats sampled (again, an inset zoom would help here).

Thank you for this comment. We have now added an example of each urban and rural sampling point in Figure 1a (based on Stamen design), so the reader can get a visual idea on the urbanisation intensity on each population. We already provided a general description on the sampling sites and their beforehand choice on L473-482, but in this new version we have also added a new figure with the proportions of each’s site landcover to further exemplify the habitat differences (Fig S1). Nonetheless, birds are really mobile endotherm organisms, and, in this study, we ignore the precise spatial use of the sampled individuals; therefore, we preferred to avoid adding excessive abiotic or biotic information about the populations as this could be misleading and not the aim of the study.

Line 47: this isn’t entirely accurate. There have been a handful of studies examining convergence in different cities for a variety of taxa, including plants, lizards, fish, and invertebrates. If the authors specifically mean genetic convergence and not phenotypic convergence, then this should be stated. Even so, there are still examples of adaptive genetic convergence in multiple urban populations (thinking specifically of three examples: fish, lizards, clover) that should be acknowledged.

We are sorry for the oversight and have revised this whole section of the manuscript accordingly and added more references to exemplify it (L48-53).

Line 89-92: I don’t think your data support this conclusion. The urban and rural separate for

Lisbon, and maybe Glasgow, but the other 7 populations are clustered together with no clear separation. Your population structure data suggest a panmictic population not independent colonizations. A haplotype map would help resolve population level relationships. If these truly are not independent colonizations then the conclusion of parallel selection is invalid.

Thank you for pointing this out. After the new population structure analyses, we believe that is possible to consider 8-9 population pairs (see response to *Reviewer #1* for details). We have also included new figures in the current version, including a map of effective migration surfaces (Fig. 2e) and a *UMAP* projection (Fig. 2a), to make this conclusion clearer.

Line 114: I'm not sure I follow why larger chromosomes having more urban-associated SNPs highlights the polygenic nature of urban adaptation. Please explain.

Under a polygenic scenario of adaptation, one expects that larger chromosomes (or chromosomes with more genes), which theoretically contain more mutational targets, explain a larger amount of the variance e.g., contain more significantly associated SNPs (see e.g., Santure et al. 2015, *Mol Ecol.*).

Line 117: variance in PC1 is only 1.98% ? That seems very low to be meaningful.

We agree that the variance explained by PC1 is very small, but that is somehow expected for a species with large population sizes, large-scale dispersal and high genomic diversity (standing genetic variation). Similar patterns can be found for example in coyote (*Canis latrans*), see Heppenheimer et al. 2018 *Ecol. Evol.*

In our study, because of the fact that all individuals are genetically very variable and the differentiation between urban and rural populations does not always affect the same genomic regions, we do not find a stronger genome-wide trend along PC1 or PC2 (i.e., stronger signals and more variance explained). However, to summarise a larger amount of genetic variation in 2 dimensions, we include in the revised version a *UMAP* projection (Figs. 2a and S3 b and c), which more effectively displays the genetic variation in this system.

Line 120-121: Why are these shared SNPs likely involved in local urban adaptation? Other non-adaptive processes could result in these patterns.

Thank you for pointing this out. In the revised version we have performed further analyses to discard other evolutionary drivers of genetic differentiation (see Results and discussion: "*Evolutionary drivers of genetic differentiation and signatures of selection*"). The new analysis point towards a general role of divergent selection over background selection or drift. Moreover, the fact that these SNPs are detected by two independent methods (*LFMM* and *BayPass*), one of them which also correlates with the direction of local adaptation (i.e., *BayPass*, Forester et al 2018 *Mol. Ecol.*) give us a high certainty on their probable role in the urban adaptation. Also, almost half of the genes associated with them are detected under selection in urban populations.

Line 124: Please explain this analysis / model. What are the terms? What is the purpose? Which urban-associated SNPs comprise PC1 (the core SNPs)? Isn't this kind of circular — identifying SNPs associated with urbanization then asking if they differ by habitat type (urban v non-urban)?

We apologise for not being clearer before in this section. We have now explained the linear model and corresponding terms in more detail (see L199-206; and *Methods* section: L650-660). The purpose of this analysis is to identify how similar allele frequency differences at urbanisation-associated SNPs are across all urban-rural pairs. While the GEA analyses identify SNPs associated with urbanisation across localities (PC_{urb}), these do not have to necessarily diverge in the same direction in all populations or diverge to a different extent (as confirmed by our analysis). Detecting an association in a heterogeneous meta-population, even with strong gene flow and weak differentiation as it is the case in our study, does not necessarily mean that the strength and direction of the obtained association is the same in all populations. Therefore, these analyses try to further test the similarity in the detected allele frequency differences.

In the previous version we performed the analysis based on PC1 scores for all *LFMM-SNPs* and “*core-urbanisation SNPs*” (see L213-224 in the new version). However, in the revised version we have also included the same analysis on a SNP-by-SNP basis as well (all *LFMM* and “*core-urbanisation SNPs*”) in order to also confirm this pattern on an individual SNP basis (L206-212; Figs. 3d and S6b). The results are very similar and indicate that there is variation in how similar allele frequency differences are, although the pattern for the “*core-urbanisation SNPs*” show a stronger similar allele frequency shifts across all localities.

Figure 2: It would be informative if the “core urban SNPs” were indicated in the Manhattan plots (a & b).

We have now indicated the “*core-urbanisation SNPs*” in the revised GWAS plot following this suggestion (Fig. 3a).

Line 174-176: Why was Lisbon used “as the outgroup for estimating PBS”? How sensitive is this analysis to this choice of outgroup? Unclear what the correlation test presented here means, please explain.

Lisbon was initially used as the outgroup as it was genetically most distinct. However, as we now have included *XP-nSL* as an additional statistic to test for ongoing/recent selective sweeps, which is not sensitive to the outgroup of choice and more readily comparable across populations, we have removed the *PBS* analysis from the revised version. However, the general results of this analyses are largely comparable and differences in the detected genes are mostly attributed to the revised choice of threshold, as we now identify genomic windows under selection based on the mean score and the proportion of significant outlier SNPs within the respective window (see *Methods* section: “*Haplotype-based selection analyses*”).

Line 185-186: How was “elevated PBS values” determined? Is there a statistical test associated with this? Please clarify.

Elevated *PBS* was determined based on the 95th percentile of the genome-wide distribution. However, we have now removed the *PBS* analysis from the revised manuscript and this comment is no longer applicable.

Line 394: Why was the scale of 1km² chosen? Is this a biologically meaningful scale?

In line with the response to the first “*Line by line comment*”, birds are extremely mobile organisms and it is difficult to predict the actual spatial use of an individual as this

will differ between life stages but also seasonally. For instance, the breeding territory size of a great tit is approx. <2 ha (e.g., Krebs 1971 Ecol.), but the size might differ between habitats, years, breeding density, etc. Therefore, we opted to use 1km² around the sampling sites in order to gain a bit of certainty on covering individuals' territory but also to ensure that that we considered the heterogeneity within each selected site. However, we cannot disregard that the sampled individuals made use of further areas within each population.

Nonetheless, we still have a high degree of certainty that the studied birds can be considered "urban dwellers" based on the season when they were sampled or the ring information and the age, as adult birds are generally residents and they usually stay close to their breeding sites. In any case, the idea was to obtain a metric to exemplify that the chosen areas could be regarded as "urban" and contrasted with the selected "rural" ones and that were representative of each urban centre.

Line 444: How sensitive is this analysis to the choice of the number of latent factors (i.e., other than the two tested)? I don't follow why the authors chose to test (only) 2 and 4 based on the referenced supplementary materials (it seems that K could also reasonably be 1 or 3).

In general, the analysis is not very strongly affected by the choice of latent factors as the population structuring in this system is generally weak (e.g., each PC only explains a relatively small proportion of variance), which in general increases the accuracy of the genotype-environment association. As the analyses with latent factors of 2 and 4 were identical, we have not tested further numbers, as these will most likely lead to the same results in the presence of such weak structuring.

Reviewer #4 (Remarks to the Author):

The manuscript compares urban and rural populations of great tits to determine whether there are repeated signatures of selection to an urban environment. The manuscript looks for concurrent shifts in allele frequencies among populations as well as selective sweeps. In addition to the interesting question of adaptation to urban environments, which is presumably rapid considering the timeframe during which species adapt, there is large interest in understanding convergent evolution. The findings are potentially novel but not currently presented in a way that is easy to interpret. Moreover, the generalizability of the findings is missing (see comment below about the goals of the paper).

We thank the reviewer for the very constructive review that helped us to increase the clarity of our manuscript. Please find our detailed responses below.

Abstract

The abstract is unclear in whether the replicability is within or between species. Moreover, the two statements about repeated polygenic responses and selective sweeps within the same genes does not jive with the final sentence that adaptation to urbanization has occurred in the same pathways. If selection was occurring in the same genes, it is a much stronger conclusion than same pathways. Therefore, the abstract is a bit misleading about the findings.

We agree that the abstract was likely not as clear as it could have been, and we have revised it substantially to highlight that the replication is within species and that we find parallelism on the gene level.

Introduction

In the introduction, it would be helpful for the manuscript to include information about how species have phenotypically adapted, as there are examples of the challenges but no evidence of the actual phenotypic adaptations that have occurred, especially because “these adaptations” are referred to (lines 38-41).

We have now revised the introduction to include more explicit examples of phenotypic adaptations in great tits and other species to life in urban habitats (L37-42), which e.g., include changes in behaviour such as flight initiation distance, cognitive skills or other behavioural syndromes in multiple birds’ species (see e.g., Sol et al 2013 Anim. Behav. for an early revision on the topic).

There is a substantial debate about convergence vs parallelism and it would be helpful for the manuscript to define their use of the terms, as parallel and convergent are used seemingly interchangeably.

We apologise for the unclarity and acknowledge the ongoing debate about convergence vs parallelism. In general, we think that these terms can be used interchangeable in some situations, but we also agree that the use of “parallelism” is more appropriate when studying recent cases of divergence and replicated evolution, as in our study. Therefore, we have revised the text in the manuscript to refer to parallelism and have defined the use of terminology in the introduction (L48-64; aims of the study: L71-77).

A question that occurred to me in the introduction is what is the site fidelity for the species?

While this is a genomic study, it is important to include this essential information about the study species in the main text, and not solely in the methods. (for example, information around line 378). Also, what is the generation time of great tits?

We agree on the relevance of this information and as previously indicated to Reviewer #3, we have a high degree of certainty that the studied birds can be considered “urban dwellers” based on the season when they were sampled or the ring information and the age, as adult birds are generally residents and they usually stay close to their breeding territories. However, we cannot ensure that they made use of further areas. We have now included some information on territory size in the *Methods* section (L497-500) but still further research is needed in urban areas as this could differ from other populations.

Great tit generation time is around 2 years (e.g. Garant et al 2004 Am. Nat., Gienapp et al 2013 Proc. Royal. Soc. B), but we cannot exclude that this differs between habitats or populations and thus we preferred to not mention or use this information in our analyses or discussion.

The goal of the study is not entirely clear, in the abstract and most of the introduction, the manuscript is painted as one studying convergent evolution but at the end of the introduction the manuscript states that the study provides “an outline of genes to target in future ... studies”. I am left with wondering whether the goal is to identify genes to target or a more evolutionary inference endeavor. Of course, these can both be goals of the manuscript but as it is written, the goal(s) are not clear and not written as complementary.

We apologise for the lack of clarity in the study goals. We have now clearly stated the main goal of the manuscript in the introduction, which is to “...identify the genetic basis of urban adaptation and test its repeatability...” (L66-67). Also, we have now clearly stated the core objectives of the study and the approaches we have followed to address these (L73-77), together with the complementary value of providing an outline of genes to target in future ecological and functional studies (L77-80).

Line 82, the statement “slightly lower” has no statistical support ($P=0.377$) and should be removed.

We agree and this statement has been now removed from the current version.

PCA does not show that pairs are independently colonizing urban and rural habitats. The TreeMix tree would be much more informative with the rural pairs, which would actually show independent colonization and closest pair. There is sufficient data for each population to include in the TreeMix tree.

We agree that it is more informative to split every locality by urban and rural, particularly due to the presence of genetic differentiation. We have therefore performed the *TreeMix* as suggested by the reviewer. Similar to the new *UMAP* and *EEMS* population structure analyses (Fig. 2), the revised *TreeMix* tree suggests the independent and repeated divergence of urban and rural populations in all localities, except in Munich and Paris, for which the separation is less pronounced. This is supported by a lack of separation in the *UMAP* plot (Fig. 2a) and estimates of increased gene flow between these two cities in the *EEMS* analysis (Fig. 2e). Thus, in all localities, except Munich and Paris, we are relatively certain that urban adaptations occurred separately. However, as genomic landscapes of differentiation between urban and rural populations differ between Munich

and Paris (e.g., Figs. 4a and S11), we hypothesize that urban adaptations still occurred separately but from a more strongly admixed rural (meta-) population.

What was the “control” for LFMM? The 1% FDR cutoff is set by the user and the wide distribution across the genome plus the larger number of chromosomes containing a larger number of SNPs suggests to me just random hits. When imposing a user-selected cutoff, there is always a tail. Perhaps a randomization of the population assignments (PCurb) and re-running to show that the findings are unexpected given the incorrect assignment.

In GEA analyses of non-model species the choice of adequate independent controls is not possible. However, as suggested by the reviewer, we have performed randomisation analyses, permutating population assignments 20 times and re-running LFMM, and used the 95th percentile of the random distribution of Z-scores to define a significance threshold (e.g., Fuller et al. 2020 Science) (Fig. S4a; L563-572). Yet, we found this approach to be more liberal than the 1% FDR threshold, resulting in more significantly associated SNPs (4,358 vs 2,758 ; L568-570). Thus, we retained the more conservative FDR-based detected SNPs for further downstream analyses.

Furthermore, our LFMM analysis detect a significant overlap with the more conservative BayPass approach, which strongly corrects for population structure. This result suggests that our associations are likely true positives. In addition, the fact that the significantly associated SNPs (LFMM and “core urbanisation SNPs”) strongly distinguish urban and rural populations across Europe (which is not the case for the full dataset), together with the parallel allele frequency differences in most of the cases, further points towards true-positives and not random associations.

If 2 SNP sets are drawn from the full SNP set, what is the chance they will overlap by 34 SNPs?

A resampling-based permutation approach shows that 34 SNPs represent significantly more overlap than expected ($\chi^2 = 31.2$, $P = 2.347e-08$; L179-180).

Why are the urbanization-associated SNPs indicated with GWAS? GEA may be more appropriate but all of the subscripts on the PCs is a bit unwieldy.

We agree and have revised it throughout the manuscript as GEA.

Figure 2f is confusing because minor allele frequency is on the y-axis, yet which population is the MAF measured in? Both populations have MAF > 0.5, which is impossible.

We agree that Fig. 2f in the previous version of the manuscript was not intuitive as MAF was the mean across all urban and rural populations. Therefore, we have revised this part and included figures with linear model effect sizes for each associated SNP (Figs. 3d, S6 b and d).

How was the overlap in genomic windows determined? (line 164-165)

We determined the overlap of genomic windows using the *plyranges* R-package, selecting windows that overlap by any number of base pairs.

The title for Figure 3 is a bit misleading because the figure does not show “shared signals of selection”, it shows signals of selection, some of which may be shared.

We agree, and we have revised the previous title and re-write a more appropriate one for Fig. 4 (formerly Fig.3): “Signatures of selection in urban populations”.

What is the relevance of the highly negative PBS values? Are those regions that are shared among the rural landscapes?

Highly negative PBS values are technically regions shared across rural populations but effectively zero along the target branch, i.e., in the respective urban population. However, as we have included new statistics of ongoing/recent selection, i.e., *XP-nSL*, in the revised manuscript, we have excluded PBS in the current version, as it is more likely to be sensitive to the choice of outgroup. *XP-nSL* does not require an outgroup population but only the urban and adjacent rural population.

Were the outliers on the Z-chromosome chosen with the same 99% cutoff as the rest of the genome?

In the original manuscript, outliers on the Z-chromosome were chosen with the same genome-wide cut-off as for autosomes. However, we have revised this, and implemented separate 99th-percentile cut-offs for the Z-chromosomes and the autosomes, based on separate distributions (L617-627; Fig. 4a).

Line 307, the 79 genes were not detailed in the LFMM analysis, however 79 genes were mentioned in the PBS / Fst overlap among 3+ populations.

This section has been removed from the revised manuscript.

What is the evidence for recurrent but independent selective sweeps? Same gene but independently in each urban population?

We are sorry for not being clear here before. The hypothesis that recurrent sweeps in the same gene occurred independently comes from the observation that sweep signatures and haplotypes are not shared across most populations, meaning different genomic windows that overlap with the same gene. For example, Figs. 5a and b show signs of selective sweeps, but sweep haplotypes are not shared across urban populations (Fig. 5b). Also, the sharing of genes with sweep signatures is higher than the number of shared sweep windows (e.g., comparing Figs. S10 and S15).

How was the number of sex chromosomes accounted for in the analyses? I did not see a separate analysis of the sex chromosomes in the methods but perhaps I missed it.

Thank you for this comment. Indeed, this is an important factor to consider and we have now re-run the analysis including and excluding sex chromosomes, as well as accounting for potentially different patterns on the sex chromosomes (see L617-627). We acknowledge that genotyping on the sex chromosome can be problematic using whole-genome sequencing approaches when the coverage threshold is too low and does not account for the differences in the number of sex chromosomes between males and females (e.g., Van Belleghem et al. 2017, Mol. Ecol.). However, the stringent filtering and genotyping accuracy of the Affymetrix SNP-chip allows for accurate genotyping on the sex-chromosome as it is less reliant on coverage.

Why was a 1km by 1 km rectangle chosen and how does the 1 km x 1km rectangular area related to bird behavior? (line 394).

Thank you for pointing this out. We have already indicated the rationale for this to *Reviewer #3*, birds are extremely mobile organisms and it is really difficult to predict the actual spatial use of an individual as this will differ between life stages but also seasonally. The relevance on the use of a 1km² area around the sampling sites has been previously validated across avian studies (e.g., Vincze et al 2017 *Front. Ecol. Evol.*) and basically aims to gain a bit of certainty on covering individuals' territory but also to ensure that that we considered the heterogeneity within each selected site. However, we cannot disregard that the sampled individuals made use of further areas within each population or that the habitat structure has changed within the studied individuals' lifetime, in particular in the urban populations. In any case, this metric only intends to provide a proxy for the human impact in each population as assessing the impact of the urbanisation on each particular individual is an almost impossible task to achieve in natural populations.

Fst calculations are described twice in the methods with two different approaches, which was actually used? If two different estimators are used in two different programs, why? And which estimator is used by *SNPRelate*?

We are sorry for the lack of clarity and we have now revised the methods. For all the revised analyses we are using the Weir-and-Cockerham's F_{ST} estimates (in windows and genome-wide) calculated with *VCFtools* (L589-595).

What LD value was used for pruning based on LD with *plink*? (line 427) some methods details are missing.

We have overall revised the *Methods* section to include more detail on this respect. Regarding the LD pruning, we have used the following settings in *Plink 1.9*: “-*indep 50 5 2*” (L536-537).

The number of SNPs retained and then reported in the filtered dataset differ (line 428).

We are sorry for the mistake and have corrected this throughout the manuscript.

Do the polygenic findings overlap with the selective sweeps?

Yes, in some cases the polygenic findings of the GEA overlap with shared selective sweeps. In the revised manuscript we have tried to make this clearer throughout the text (e.g., L375-379) and also e.g., Fig. 5a and supplementary Table S3.

REVIEWER COMMENTS

Reviewer #1 (Remarks to the Author):

I have now found the time to look at the revision of this paper, and I am happy to say that the authors have responded adequately to most of my questions. I would like to congratulate them on this revision. Also, I am sorry to have taken that long, but the work load of the end of last and beginning of this year has prevented me from handing in a report earlier.

I would like to point out again to the editor and also to the authors that given the recent events around Nature journals and in Nature Communications, I only accepted reviewing this manuscript for Nature Communications as a last-time exception; out of respect for the authors' work and because I deem a change of reviewers mid-process unfair.

I have only a number of minor comments left:

1. To my understanding the abstract still puts a strong emphasis on parallel evolution. However, as it seem to me and as was also commented on by Reviewer 3, the results would rather reflect a mix of parallel and idiosyncratic responses to urban selection pressures. I think this could and should be worked out still better.
2. L31: Is it really the same genes? Or genes in the same pathway/with similar function?
3. L53: Delete THE from „the urban adaptation“
4. L72-73: „... we analysed nine paired urban and rural great tit populations...“ > please rephrase: « ... we analysed pairs of urban and rural great tit populations from nine localities. » This better reflects that you did not know whether they are differentiated populations when starting out.
5. L97: „per site“ >> „per SAMPLING site“
6. L112-114: I would tone down on the support through TreeMix. Ultimately, both reflects a deficit in heterozygosity and is based on the same signal. It is thus not independent confirmation.
7. L227-228 : Only part of the adaptation is parallel. See comment 1.
8. For the haplotype-statistics analyses, the authors only report the number of windows that are outliers. However, what would be more interesting is how many regions are outliers and maybe how big they are. That is, I would suggest that adjacent windows are counted as a single region and this is reported in addition of (or instead of) the number of windows.
9. L284: „Landscape of selection“ > This is too affirmative of selection as the only process generating these patterns. I would suggest to rather refer to the pattern than to a particular process driving it.

10. L332 : It does not necessarily have to be background selection. Other types of LINKED selection could also explain it. I think it is important to make this distinction here and elsewhere.

Reviewer #3 (Remarks to the Author):

I have completed reviewing the revised manuscript by Salmón et al., “Repeated genomic signatures of adaptation to urbanisation in a songbird across Europe”. In my previous review, I raised three key concerns regarding the sequencing methods, population structure, and non-adaptive processes, in addition to several minor questions.

I found this manuscript to be substantially improved by the review process and much easier to follow in this revision. There are many things about this manuscript that I really like, and I think it will be an important contribution to the literature. The continent-wide scale is impressive and is not matched by any urban evolutionary genomics study I can think of, and the authors do a nice job of tying together several analyses to paint a picture of urban evolutionary change at the genomic level. I am satisfied with the thorough responses and revisions with respect to my first and third concern and appreciate the detailed responses to my line by line comments. I am convinced that the authors have detected genomic signatures of natural selection in response to urbanization in this species across Europe. However, a main concern from my previous review and one echoed by Reviewers 1 and 4 remains – I am not convinced that the patterns of repeated genomic signatures can be conclusively attributed to parallel natural selection given the lack of population structure and evidence of strong gene flow.

I, along with Reviewers 1 and 4, previously raised concerns over the lack of apparent population structure. I do not feel that these concerns have been remedied in this revision, although I appreciate the authors’ additional analyses and attempts to resolve this issue.

I agree that there is some population structure, as the authors point out, but the fastStructure analysis found 4, not 8 or 9 populations. Moreover, I do not see much additional clarification with the UMAP analysis compared to the previous PCA (there is still a dense cluster in the center – perhaps a zoomed inset of this would help?). I might suggest that the authors see what population structure looks like with DAPC, which should maximize the between group variance, but I suspect that a DAPC will mirror the PCA findings of minimal population structure.

The new TreeMix analysis, which was suggested by Reviewer 4, is a step in the right direction, but I don’t think it is strong enough as is to conclude independent colonizations between urban and rural pairs. TreeMix allows for incorporation of m number of migration events. I suggest the authors repeat the TreeMix analysis with varying numbers of migration events (it is unclear if any migration was allowed in the model presented). This might help resolve if migration is widespread (i.e., if the best supported tree

is one with a large number for m) or rare (e.g., only between Munich and Paris). If the best supported tree does support only one or two migration events, this would be strong support for independent colonization.

As presented, the population structure analyses consistently point to a single conclusion – one of extensive gene flow with little population differentiation and in which an urban population is not necessarily most closely related to its rural pair. Unfortunately, to me, this is a hard stop for the parallelism narrative. Ideally, to set up the narrative of parallelism one would want to see each urban population most closely related to the rural pair, and for F_{ST} to be smaller between urban-rural pairs than between different regions (and ideally large F_{ST} values between regions). The PCA/UMAP analysis does not clearly show urban-rural paired clusters and the F_{ST} values suggest panmixia across the range. Simply finding that gene flow is slightly reduced regionally (EEMS analysis) or between urban compared to rural sites (F_{ST} analysis) is not sufficient – these analyses still suggest ongoing gene flow.

In addition to this main concern, I have four minor comments:

(1) LFMM is sensitive to the number of clusters provided. The authors specify $k=2$ and $k=4$ based on Admixture results, but the Admixture analysis was removed from this version. The authors should update the text to specify that the choice of k is supported by their new population structure analyses. They should also provide a histogram of the p -values (or qqplot) as recommended in the manual for LFMM to verify that the choice of k has not resulted in overly conservative or liberal results.

(2) EEMS is sensitive to the number of demes specified by the researcher and the EEMS manual recommends running the analysis with multiple choices for demes. I suggest the authors rerun the EEMS analysis with varying deme numbers as recommended to be confident in the gene flow patterns they have presented, or otherwise provide justification for their choice of 1000.

(3) Lines 135-137 “most urban populations aggregated along independent genetic clusters together with their adjacent rural counterparts” – I don’t see how this is accurate. The large cluster in the middle suggests extensive genetic exchange between those populations and other pairs such as Lisbon, Milan, and Malmö have urban populations more closely related to other populations than they are to their paired rural population. I don’t see how the authors can state that urban populations pair with their respective rural populations, at best this is the case for only Glasgow, Barcelona, Gothenburg, and Madrid.

(4) Lines 260-263 “The number of shared outlier windows between urban populations was not correlated with their genetic or geographic distance” – OK, this helps convince me that the results could have independently arisen, but what about the identity of the outlier windows? In other words, do the outlier windows tend to be common among geographic neighbors? One can imagine a situation where the number of outlier windows shared with at least one other population is high regardless of geographic distance, but in which geographically close populations tend to have similar sets of shared windows but the same overall number – e.g., instead of loci ABCD shared across all 9 populations, ABCD

are shared between population 1 and 2, EFGH between 2 and 3, IJKL between 3 and 4 etc. – in this scenario each population shares 4 loci with at least one other population but the identity of shared loci differs. If there is a geographic pattern to the identity (not number) of shared loci that would be counterevidence of independent selection in each city. Or perhaps I am either overthinking this or am misunderstanding the analysis. Clarification here would help.

Overall, I really like this manuscript and I am very excited by the results, I just don't think the parallelism narrative is well-enough supported to be the main focus. The authors do detect compelling signatures of selection and their results are intriguing. However, given the lack of clear population structure even for the most divergent population pairs (F_{ST} analyses suggest panmixia) I think they go too far to use their dataset as a test of parallelism. I think the study is just as intriguing as a story of urban adaptation at different levels of biological hierarchy and continent-wide urban sweeps, which is novel in urban genomic research, without strongly emphasizing parallelism. Even if loci are transferred between sites because of migration, they are still maintained at high frequency in the urban habitats compared to nearby rural habitats, suggesting they are important for urban population persistence and are likely targets of urban natural selection.

In the following document, we present point-by-point response to each reviewer's critique (*italics* typeface), with our responses in **bold** typeface. Note that the line numbers in the responses refer to those in the new version of the manuscript without track changes.

Reviewer #1 (Remarks to the Author):

I have now found the time to look at the revision of this paper, and I am happy to say that the authors have responded adequately to most of my questions. I would like to congratulate them on this revision. Also, I am sorry to have taken that long, but the work load of the end of last and beginning of this year has prevented me from handing in a report earlier.

I would like to point out again to the editor and also to the authors that given the recent events around Nature journals and in Nature Communications, I only accepted reviewing this manuscript for Nature Communications as a last-time exception; out of respect for the authors' work and because I deem a change of reviewers mid-process unfair.

We really appreciate your time and thoughtful and constructive comments (both in the initial revision and in the current one), which we believe have enormously improved the manuscript. Please find our detailed responses below.

I have only a number of minor comments left:

1. To my understanding the abstract still puts a strong emphasis on parallel evolution. However, as it seem to me and as was also commented on by Reviewer 3, the results would rather reflect a mix of parallel and idiosyncratic responses to urban selection pressures. I think this could and should be worked out still better.

We agree and have made it more clear in the abstract and throughout the manuscript that genomic responses most likely reflect a mix of repeated selection and other processes (e.g., drift, gene flow or linked selection), e.g., L32, 50, 53-54, 56-61, 70-73, 165-166, 182, 207-234, 235-239, 253, 373, 375-378, 407-408, 484-487, 681.

In addition, we have modified the title from: "*Repeated genomic signatures of adaptation to urbanisation in a songbird across Europe*" to "*Continent-wide genomic signatures of adaptation to urbanisation in a songbird*". We believe that this change in the title helps to avoid any misunderstanding regarding parallelism and is in accordance with Reviewer's #3 comment: "... I think the study is just as intriguing as a story of urban adaptation at different levels of biological hierarchy and continent-wide urban sweeps...".

2. L31: Is it really the same genes? Or genes in the same pathway/with similar function?

We refer to the same genes.

3. L53: Delete THE from „the urban adaptation“

We have now deleted the word "the" as suggested.

4. L72-73: „... we analysed nine paired urban and rural great tit populations...“ > please rephrase: « ... we analysed pairs of urban and rural great tit populations from nine localities. » This better reflects that you did not know whether they are differentiated populations when starting out.

We agree with the suggested rephrase and the phrase has now been edited accordingly (L69-70).

5. L97: „per site“ >> „per SAMPLING site“

Thank you we have now added the suggested word in the main text (L93).

6. L112-114: I would tone down on the support through TreeMix. Ultimately, both reflects a deficit in heterozygosity and is based on the same signal. It is thus not independent confirmation.

We agree and have rephrased this sentence to make this clear (L110-111).

7. L227-228 : Only part of the adaptation is parallel. See comment 1.

Thank you and we agree. Therefore, and in accordance with *comment #1* we have modified the two initial sentences of the paragraph in order to minimise any claim on parallelism: “Together, we show that there is local adaptation to urban habitats in great tits, and that this occurred across Europe through shifts in allele frequency of the same loci. This might have occurred via the standing genetic variation observed for the species, as putatively adaptive alleles were shared across large parts of the species' European distribution” (L235-238).

8. For the haplotype-statistics analyses, the authors only report the number of windows that are outliers. However, what would be more interesting is how many regions are outliers and maybe how big they are. That is, I would suggest that adjacent windows are counted as a single region and this is reported in addition of (or instead of) the number of windows.

We apologise for this and we have now also included the number and size of the genomic windows detected as outliers in each of the analyses, see e.g., *XP-nSL* (L258-261) or *Rsb* (L309-312).

9. L284: „Landscape of selection“ > This is too affirmative of selection as the only process generating these patterns. I would suggest to rather refer to the pattern than to a particular process driving it.

We agree with the reviewer's suggestion and we have exchanged “Landscape of selection” for “variation in XP-nSL”. Here is the change within the context of the referred sentence: “The observed landscape of variation in XP-nSL is in agreement with the genomic landscape of genetic differentiation (standardised Z-transformed F_{ST} [ZF_{ST}]) between each adjacent urban and rural pair across the same windows...” (L300).

10. L332 : It does not necessarily have to be background selection. Other types of LINKED selection could also explain it. I think it is important to make this distinction here and elsewhere.

We agree and have revised the manuscript accordingly, referring to linked selection rather than background selection, e.g., L350-351, 353, 365-370.

Reviewer #3 (Remarks to the Author):

I have completed reviewing the revised manuscript by Salmón et al., “Repeated genomic signatures of adaptation to urbanisation in a songbird across Europe”. In my previous review, I raised three key concerns regarding the sequencing methods, population structure, and non-adaptive processes, in addition to several minor questions.

I found this manuscript to be substantially improved by the review process and much easier to follow in this revision. There are many things about this manuscript that I really like, and I think it will be an important contribution to the literature. The continent-wide scale is impressive and is not matched by any urban evolutionary genomics study I can think of, and the authors do a nice job of tying together several analyses to paint a picture of urban evolutionary change at the genomic level. I am satisfied with the thorough responses and revisions with respect to my first and third concern and appreciate the detailed responses to my line by line comments. I am convinced that the authors have detected genomic signatures of natural selection in response to urbanization in this species across Europe. However, a main concern from my previous review and one echoed by Reviewers 1 and 4 remains – I am not convinced that the patterns of repeated genomic signatures can be conclusively attributed to parallel natural selection given the lack of population structure and evidence of strong gene flow.

Thank you for your constructive review and for taking the time for checking our previous responses to Reviewer’s #2 and #4 comments and criticism. We are pleased to hear that you find the new version of the manuscript improved and that you like the idea and results behind our work. Please find our detailed responses below

I, along with Reviewers 1 and 4, previously raised concerns over the lack of apparent population structure. I do not feel that these concerns have been remedied in this revision, although I appreciate the authors’ additional analyses and attempts to resolve this issue.

I agree that there is some population structure, as the authors point out, but the fastStructure analysis found 4, not 8 or 9 populations. Moreover, I do not see much additional clarification with the UMAP analysis compared to the previous PCA (there is still a dense cluster in the center – perhaps a zoomed inset of this would help?). I might suggest that the authors see what population structure looks like with DAPC, which should maximize the between group variance, but I suspect that a DAPC will mirror the PCA findings of minimal population structure.

Thank you for the suggestion and we have now run a DAPC based on approx. 21,000 unlinked SNPs in *adegenet*. The DAPC revealed slightly stronger population structure than the PCA/UMAP analyses (DAPC plot below for DA1 vs DA2). As is possible to observe in the figure, the DAPC analysis separates all individuals into distinct clusters by sampling location, i.e., urban and rural. However, similar to the results obtained with UMAP/PCA, it does not show consistency in clustering all urban and rural pairs. Moreover, Lisbon populations (LIS) still present the highest distinction among the studied populations (see DA1 in the figure below). This result is in agreement with some other analysis included in the current version, which already suggest the presence of some albeit weak population structure, e.g., significant pairwise F_{ST} results (Table S2) or presence of isolation-by-distance (Fig. 2d). Therefore, we have taken the decision of not presenting the DAPC in the manuscript or supplementary text as they do not add much to the interpretation of the data and are in line with other results reported in the text. We hope the reviewer agrees on this decision, but if the reviewer still thinks that it is valuable to include, we are of course willing to do so.

Figure: DAPC with DA1 (x-axis) vs DA2 (y-axis) based on ~21k unlinked SNPs.

The new TreeMix analysis, which was suggested by Reviewer 4, is a step in the right direction, but I don't think it is strong enough as is to conclude independent colonizations between urban and rural pairs. TreeMix allows for incorporation of m number of migration events. I suggest the authors repeat the TreeMix analysis with varying numbers of migration events (it is unclear if any migration was allowed in the model presented). This might help resolve if migration is widespread (i.e., if the best supported tree is one with a large number for m) or rare (e.g., only between Munich and Paris). If the best supported tree does support only one or two migration events, this would be strong support for independent colonization.

Thank you for this comment and suggestion and we apologise for not having been

clearer on the *TreeMix* analysis in the previous version. We initially ran *TreeMix* with “ $m=0$ ” up to “ $m=5$ ” migration events, but it was our oversight not including these results in the manuscript or supplementary. Determining the optimal number of fitted migration events is not straightforward using one metric. Therefore, for obtaining this number we combined the likelihood estimates for each tree, i.e., the total variance explained by each tree with increasing numbers of migration

events (estimated using the following script:

<https://github.com/darencard/RADpipe/blob/master/treemixVarianceExplained.R>),

together with the significance of fitted migration events (p-value included in the *.treeout file). Based on these estimates (L571-574), we concluded that more than 2 migration events do not significantly improve the fit of the tree. This decision is based on: *i*) the Variance Explained starts plateauing at “m=2” (Fig. S4a, attached below); *ii*) From “m=3” it starts fitting non-significant migration events ($p > 0.05$); and *iii*) the likelihood also shows a local plateau, although it increases afterwards (Fig. S4a). The significant migration events that are fitted in the tree with “m=2” are from Lisbon (LIS) to Glasgow (GLA) and from Malmö (MAL) to Madrid (MAD) (see Fig. S4c). These results are included now in the revised supplementary material, including trees with fewer or more migration events (“m=1” and “m=3”).

Overall, fitting migration events slightly increased the fit of the tree but fitted migration mostly seem historical and not specific e.g., between urban populations. While we still think that some urban populations repeatedly and independently colonized urban habitats, as multiple urban populations form distinct genetic clusters (e.g., Lisbon -LIS-, Glasgow -GLA-, Madrid -MAD- or Milan -MIL-), we cannot fully exclude that gene flow between populations facilitated urban adaptation through the spread of adaptive alleles. In the current version we have revised the text accordingly in order to provide a more balanced interpretation of the population structure results (L101-105, 110-111, 132-138, 159-166; Fig. S4).

Fig S4. To facilitate the review process, we present a copy of this figure here (see Supplementary Material file for details)

As presented, the population structure analyses consistently point to a single conclusion – one of extensive gene flow with little population differentiation and in which an urban population is not necessarily most closely related to its rural pair. Unfortunately, to me, this

is a hard stop for the parallelism narrative. Ideally, to set up the narrative of parallelism one would want to see each urban population most closely related to the rural pair, and for F_{ST} to be smaller between urban-rural pairs than between different regions (and ideally large F_{ST} values between regions). The PCA/UMAP analysis does not clearly show urban-rural paired clusters and the F_{ST} values suggest panmixia across the range. Simply finding that gene flow is slightly reduced regionally (EEMS analysis) or between urban compared to rural sites (F_{ST} analysis) is not sufficient – these analyses still suggest ongoing gene flow.

After revisiting our analyses, we agree with the reviewer's comment. We have now revised the presentation of the results based on the advice. In the new revised version, we have emphasised the narrative on the geographic variability of genomic responses to selection across urban localities on a continent-wide scale and discuss in a more balanced way the potential role of gene flow in facilitating urban adaptation (e.g., L159-166, 235-239, 278-280, 368-370).

In addition to this main concern, I have four minor comments:

(1) LFMM is sensitive to the number of clusters provided. The authors specify $k=2$ and $k=4$ based on Admixture results, but the Admixture analysis was removed from this version. The authors should update the text to specify that the choice of k is supported by their new population structure analyses. They should also provide a histogram of the p -values (or qqplot) as recommended in the manual for LFMM to verify that the choice of k has not resulted in overly conservative or liberal results.

Thank you for pointing out this oversight and we have adjusted this section accordingly (L592-593) and also provide a histogram of p -values in the revised supplementary (Fig. S5a).

(2) EEMS is sensitive to the number of demes specified by the researcher and the EEMS manual recommends running the analysis with multiple choices for demes. I suggest the authors rerun the EEMS analysis with varying deme numbers as recommended to be confident in the gene flow patterns they have presented, or otherwise provide justification for their choice of 1000.

Thank you for the suggestion. We have now also tested EEMS with 200, 500 and 2,000 demes (see attached figure). Overall, the new analyses detect very similar patterns, with high migration rates in the central part of the sampling range and reduced migration in the remaining areas. However, this is not surprising, as 1,000 demes already provide a relatively fine-scale resolution in relation to our sampling design, i.e., relatively large geographic distances between populations, 5-70km. In order to gain clarity in this respect we have included this information in the revised manuscript, although we have preferred to not include the figures below to avoid too much redundancy in the supplementary information (L579-580).

Figure EEMS. Comparison of effective migration surfaces for different numbers of demes (200,500,1000,2000). Note that the exact locations of the collection points do not always match the points shown on the maps, and across maps, as points are randomly placed at deme edges. Thus, the placement of sampling locations depends on the placement of demes, which might differ between runs and is dependent on the density/number of demes used. This also leads sometimes to the splitting or combining of urban and rural populations. $nDemes = 1,000$ is the setting used for the main manuscript. In the main text, we have overlaid the actual location of each city on this map.

(3) Lines 135-137 “most urban populations aggregated along independent genetic clusters together with their adjacent rural counterparts” – I don’t see how this is accurate. The large cluster in the middle suggests extensive genetic exchange between those populations and other pairs such as Lisbon, Milan, and Malmö have urban populations more closely related to other populations than they are to their paired rural population. I don’t see how the authors can state that urban populations pair with their respective rural populations, at best this is the case for only Glasgow, Barcelona, Gothenburg, and Madrid.

We want to thank the reviewer for pointing out this oversight. After revising the manuscript, we agree that our previous statement was not accurate. We have now reworded those lines in order to reflect more accurately the results from UMAP and *TreeMix* analyses (including the new ones following the Reviewer’s advice) (L132-138).

(4) Lines 260-263 “The number of shared outlier windows between urban populations was not correlated with their genetic or geographic distance” – OK, this helps convince me that the results could have independently arisen, but what about the identity of the outlier windows? In other words, do the outlier windows tend to be common among geographic neighbors? One can imagine a situation where the number of outlier windows shared with at least one other population is high regardless of geographic distance, but in which geographically close populations tend to have similar sets of shared windows but the same overall number – e.g., instead of loci ABCD shared across all 9 populations, ABCD are shared between population

1 and 2, EFGH between 2 and 3, IJKL between 3 and 4 etc. – in this scenario each population shares 4 loci with at least one other population but the identity of shared loci differs. If there is a geographic pattern to the identity (not number) of shared loci that would be counterevidence of independent selection in each city. Or perhaps I am either overthinking this or am misunderstanding the analysis. Clarification here would help.

We apologise for a lack of clarity in this respect. We agree that the identity and not only the number of shared outlier windows is an important aspect. However, we believe our analyses already take this into consideration and the information on that respect is available in the supplementary figures S10, S12 and S14, which summarise the intersection of outlier datasets.

For example, Fig. S10 shows the intersection of XP-nSL outlier windows across all the population comparisons and indicates that Paris (PAR) and Gothenburg (GOT) share the greatest number of outlier windows (i.e., the same windows are outliers), followed by Glasgow (GLA) and Lisbon (LIS). Under the scenario the reviewer proposes, one would expect that Paris (PAR) and Munich (MUC) should have the same outlier windows as these populations are geographically and genetically closer, instead of Paris (PAR) and Gothenburg (GOT), which are more distant. Therefore, if geographically close populations have the same outlier windows that are not shared at all or weakly shared with more distant populations (e.g., Pop1-Pop2 share ABCD, Pop3-Pop4 share EFGH but Pop1-Pop3 only share A and Pop1-4 only share D), then you would still expect a correlation with geographic distance.

We have clarified this point in the revised manuscript: “...In general, the most pronounced sharing of outlier windows did not seem to occur between the geographically closest populations, as expected for shared genetic variation or adaptive introgression (see Fig. S10). For example, Gothenburg and Paris, and Lisbon and Glasgow, showed the highest number of shared outlier windows (53 and 47 respectively) ...” (L270-273). We hope this way we make clearer that geographically close populations do not necessarily show the same selective sweep signatures.

Also, more subjectively, in Fig. 5a, we show the top shared genes and the populations in which they were associated with signals of selection. It is possible to appreciate that they show a relatively patchy distribution. For example, HRT7 shows signs of selection in Glasgow (GLA), Malmö (MAL), Paris (PAR), Munich (MUC) and Madrid (MAD). If there were shared genetic variation or gene flow between geographically close populations, you would expect a more continuous pattern with e.g., signatures of selection in all Iberian populations (Barcelona, Madrid and Lisbon). We have clarified this in our revised manuscript: “...It is noteworthy that we did not detect an increase in the number of shared genes under selection in relation to geographical (Mantel’s $r = 0.27$, $p = 0.080$) or genetic distance (Mantel’s $r = 0.05$, $p = 0.480$), which is in accordance with the results regarding shared outlier windows and further supports that many selective sweeps likely occurred independently in multiple urban populations...” (L394-398).

Overall, I really like this manuscript and I am very excited by the results, I just don’t think the parallelism narrative is well-enough supported to be the main focus. The authors do detect compelling signatures of selection and their results are intriguing. However, given the lack of clear population structure even for the most divergent population pairs (FST analyses suggest panmixia) I think they go too far to use their dataset as a test of parallelism. I think

the study is just as intriguing as a story of urban adaptation at different levels of biological hierarchy and continent-wide urban sweeps, which is novel in urban genomic research, without strongly emphasizing parallelism. Even if loci are transferred between sites because of migration, they are still maintained at high frequency in the urban habitats compared to nearby rural habitats, suggesting they are important for urban population persistence and are likely targets of urban natural selection.

We want to thank again the reviewer for the interest in our study and for the constructive feed-back, which has greatly improved our manuscript. We hope that after revising the manuscript following both reviewers' recommendations and criticism, we have managed to remove the non-supported emphasis on parallelism from the previous version. In addition, we hope we had correctly addressed the other minor points stated by the reviewer.

REVIEWERS' COMMENTS

Reviewer #3 (Remarks to the Author):

Salmón et al.'s latest revision of their manuscript "Continent-wide genomic signatures of adaptation to urbanisation in a songbird across Europe" is a wonderful improvement over previous versions. I appreciate the authors' revised analyses and thorough responses to my many comments. All of my concerns from my previous reviews have now been addressed and I have no further comments. This is a fantastic study, nicely done!

Reviewer #3 (Remarks to the Author):

Salmón et al.'s latest revision of their manuscript "Continent-wide genomic signatures of adaptation to urbanisation in a songbird across Europe" is a wonderful improvement over previous versions. I appreciate the authors' revised analyses and thorough responses to my many comments. All of my concerns from my previous reviews have now been addressed and I have no further comments. This is a fantastic study, nicely done!

Thank you for all the constructive comments over the different revision rounds. We fully agree that the study has enormously improved and this would have not been possible without yours and previous reviewers' genuine input.